# A Review on Natural Fiber Reinforced Polymer Composites (NFRPC) for Sustainable Industrial Applications

**DOI:** 10.3390/polym14173698

**Published:** 2022-09-05

**Authors:** Siti Hasnah Kamarudin, Mohd Salahuddin Mohd Basri, Marwah Rayung, Falah Abu, So’bah Ahmad, Mohd Nurazzi Norizan, Syaiful Osman, Norshahida Sarifuddin, Mohd Shaiful Zaidi Mat Desa, Ummi Hani Abdullah, Intan Syafinaz Mohamed Amin Tawakkal, Luqman Chuah Abdullah

**Affiliations:** 1Department of Ecotechnology, School of Industrial Technology, Faculty of Applied Sciences, UiTM Shah Alam, Shah Alam 40450, Selangor, Malaysia; 2Department of Process and Food Engineering, Faculty of Engineering, Universiti Putra Malaysia, Serdang 43400, Selangor, Malaysia; 3Department of Chemistry, Faculty of Science and Technology, Universiti Putra Malaysia, Serdang 43400, Selangor, Malaysia; 4Smart Manufacturing Research Institute (SMRI), Universiti Teknologi MARA (UiTM), Shah Alam 40450, Selangor, Malaysia; 5Department of Food Science and Technology, School of Industrial Technology, Faculty of Applied Sciences, UiTM Shah Alam, Shah Alam 40450, Selangor, Malaysia; 6Bioresource Technology Division, School of Industrial Technology, Universiti Sains Malaysia, Penang 11800, Penang, Malaysia; 7Department of Manufacturing and Materials Engineering, International Islamic University Malaysia, Jalan Gombak, Kuala Lumpur 53100, Malaysia; 8Faculty of Chemical Engineering Technology and Process, Universiti Malaysia Pahang, Lebuhraya Tun Razak, Gambang 26300, Pahang, Malaysia; 9Department of Wood and Fiber Industries, Faculty of Forestry and Environment, Universiti Putra Malaysia, Serdang 43400, Selangor, Malaysia; 10Institute of Tropical Forestry and Forest Products, Universiti Putra Malaysia, Serdang 43400, Selangor, Malaysia; 11Department of Chemical and Environmental Engineering, Faculty of Engineering, Universiti Putra Malaysia, Serdang 43400, Selangor, Malaysia

**Keywords:** natural fiber reinforced polymer composites, biodegradable, natural fiber, polymer composite, sustainable: industrial applications, Industry 4.0

## Abstract

The depletion of petroleum-based resources and the adverse environmental problems, such as pollution, have stimulated considerable interest in the development of environmentally sustainable materials, which are composed of natural fiber–reinforced polymer composites. These materials could be tailored for a broad range of sustainable industrial applications with new surface functionalities. However, there are several challenges and drawbacks, such as composites processing production and fiber/matrix adhesion, that need to be addressed and overcome. This review could provide an overview of the technological challenges, processing techniques, characterization, properties, and potential applications of NFRPC for sustainable industrial applications. Interestingly, a roadmap for NFRPC to move into Industry 4.0 was highlighted in this review.

## 1. Introduction

Fiber–reinforced polymer composites based on synthetic fibers such as glass, Kevlar, and carbon fibers have advanced significantly in recent decades to meet the requirements of engineering applications. However, well-known environmental awareness in the direction of achieving the sustainability of manufactured goods has urged great potential in grueling more environmentally friendly materials with a focus on renewable raw materials in product design [1,2]. One of the most promising ways to reduce the use of synthetic fibers as reinforcement and filler materials in the building of polymer composites is through rapid growth in the field of natural fiber reinforced polymer composites. Natural fiber reinforced polymer composites (NFRPC), also known as natural fiber composites (NFC), have recently become extremely valuable materials. Natural fibers (such as hemp, sisal, jute, kenaf, flax, and others) are used as reinforcing material (fillers) in polymer-based matrices. Given the government’s emphasis on new environmental regulations and sustainability concepts, as well as the growing ecological, social, and economic awareness and the high cost of petroleum resources, the optimal use of natural resources has been enhanced [1,3]. The idea of using natural fibers as a reinforcement phase in composite materials has been around since ancient times when many civilizations used plant fibers as reinforcement. Natural fiber composites have been demonstrated as an environmentally friendly alternative to glass-reinforced or carbon-reinforced polymer composites. Natural fibers, in particular, not only reduce waste disposal issues but also reduce environmental pollution. Natural fibers (NFs), particularly as a reinforcement in composite materials, are a popular technology for a variety of applications, particularly in the context of sustainable materials. Natural fibers have significant advantages over traditional glass fibers, allowing them to compete in modern industrial applications. Natural-fiber reinforced composites have several advantages over synthetic-fiber reinforced composites, including renewability, less abrasiveness to equipment, biodegradability, high specific strength, low cost, noncorrosive, non-hazardous nature, and manufacturing flexibility [4,5,6]. 

Furthermore, society’s concern for the environment has steadily increased over the last few decades. As a result, the influence of consumerism on the environment is receiving more attention, resulting in the enactment of rules and restrictions by national governments and international organizations. The United Nations’ 2030 Agenda for Sustainable Development, which has been signed by over 190 countries, is a good example. In order to keep the oceans clean, there are special goals dedicated to recyclable plastic (goal 12) and eliminating the use of plastic bags (goal 14). Thus, increased sustainability in the composites industry necessitates basic and transformational research aimed at developing completely green composites [7]. Sustainable composites can be made from renewable resource-based sustainable polymers and bioplastics, as well as advanced green fibers such as lignin-based carbon fiber and nanocellulose. Natural fibers are lighter and less dense than mineral fibers, resulting in greater particular characteristics. This is of importance to the automotive and aerospace sectors, which are continually looking for ways to reduce vehicle weight. Natural fibers are very simple to handle and, unlike mineral reinforcements such as glass fibers, are not toxic to humans. Lignocellulosic fibers can be made from wood, annual plants, agroforestry waste, or as a by-product of industrial processes such as textile or paper production [8,9].

Nonetheless, there are several issues that should be investigated in order to normalize the usage of natural fibers rather than mineral fibers. When natural hydrophilic fibers are combined with hydrophobic matrices, weak interphases form, compromising the composites’ potential mechanical qualities [10,11]; this is a very active topic of research that has made significant progress. Multiple solutions must be active in order to create interphases that ensure a satisfactory combination of the materials’ tensile and impact capabilities. Furthermore, natural fibers have a greater range of inherent qualities than man-made reinforcements. Moreover, plant fibers’ biodegradability helps maintain a healthy ecology, whereas their low cost and good performance meet the industry’s economic needs. One type of natural fiber, the pineapple, *Ananas comosus* (L.) Merr., is a tropical plant native to Southeast Asia that was first brought to Tanah Melayu in 1922. It belongs to the Bromeliaceae family. According to data from the Malaysian Pineapple Industry Board (MPIB), the world’s pineapple plantation area was about 1,022,319 hectares in 2014, producing 25,439,366 metric tonnes of pineapple in that year alone. Furthermore, it revealed that Malaysia was among the top 20 countries in terms of pineapple plantation area, with 335,725 MT of pineapple produced in that year, accounting for 0.01 percent of global pineapple production. These data suggested that there is a surplus of pineapple raw material waste available in Malaysia to meet the demand of composite manufacturers, research and development departments, and other industries [12].

Meanwhile, in China, the majority of China’s bamboo is used to make houses, furniture, composite boards, flooring, and other wood-based items [13]. Bamboo businesses are also contributing to Bangladesh’s economic development, particularly in terms of harvesting, production, and marketing [14]. In Thailand, there are a large number of oil palm plantations in places such as Krabi, Suratthani, Chumphon, and other districts, totaling roughly 3250 square kilometers and producing 700,000–800,000 tonnes of raw palm oil each year. Palm oil production has been gradually increasing due to its low cost, and the industry has risen in response. Furthermore, oil palm production methods and procedures began with assessing the quality of palm fruit bunches from palm fruit. The waste oil palm residue collected from the pure oil palm comprises 12% of the oil palm bunch, which can be used for a variety of applications, including fiber manufacturing and fuel production [15]. Natural fiber reinforced polymer composites made from flax, sisal, hemp, and kenaf have a wide range of applications in the automobile sector in Germany and other industrialized and developing countries across the world. Door panels, seat backs, dashboards, package trays, head restraints, and seatback linings were all examples of automotive applications [16].

The COVID-19 pandemic encourages the widespread use of plastic PPE kits and single-use plastic items, resulting in additional sources of plastic pollution and further exacerbating the problems of marine litter and biodiversity. As a result, one of the greatest solutions to minimize the global plastic pollution problem is to use sustainable/ biodegradable plastics and composites such as NFRPC, as well as expand their usage into new fields [17,18]. Thus, this review focused on the summary of the technological challenges, processing techniques, characterization, properties, and potential applications of NFRPC for sustainable industrial applications. Additionally, as shown in Figure 1, a variety of novel industrial applications can be fully realized by utilizing the design concept of sustainability of NFRPC. 

As we move towards circular economy practices, it is crucial to have a thorough understanding of the benefits of recycling and to reprocess the material of NFRPCs to the environment as they are reused in further product applications. Each issue must be handled using the appropriate knowledge. Figure 2 illustrates the difficulties and challenges faced by non-renewable materials and how the NFRPC may overcome them by implementing the circular economy idea to produce environmentally friendly products and technology.

The findings of the current review will be a handy tool for the required stage process needed in innovation from natural fiber reinforced polymer composites for sustainable industrial applications through assisting with the selection of the best available natural fibers for sustainable composites in various functional applications. 

## 2. Properties of NFRPC, Chemical Properties, Types of NF Used in Sustainable Industrial Applications

Plant photosynthesis is estimated to produce 1000 gigatonnes of cellulose each year in the biosphere, making polysaccharides the largest organic carbon reserve on the planet [19]. Natural fibers (NFs) derived from cellulose can be divided into several categories based on the plant species and plant tissue used in their production, [20,21] such as bast (e.g., banana, flax, hemp, jute, kenaf, ramie, rattan), leaf (e.g., banana, pineapple, palm, sisal), seed and fruit (e.g., coconut, coir, cotton, rice husk), and stalk (e.g., bagasse, bamboo, barley, reed, rice, wheat, wood). Biomass from agricultural waste such as oil palm fronds [22], empty fruit bunch (EFB) [22], coir [23], straws, husks [24,25,26], sugar palm [27], water hyacinth [28], and sugarcane bagasse [29,30] are classified as natural plant fibers. Cotton, ramie, flax, bamboo, kenaf, jute, abaca, sisal, and hemp are among the fiber crops included in this category Figure 3. Plant fibers are lignocellulosic polymers composed primarily of cellulose, hemicellulose, and lignin [31].

Even when derived from the same plant, the properties of natural fiber are affected by a variety of factors, including physical characteristics, chemical composition, crystalline cellulose dimensions, microfibrillar angle, defects, structure, and isolation technique. As a result, the quality and mechanical qualities of the fiber can differ significantly. The chemical composition of common NFs is shown in Table 1.

In order to achieve optimal performance, several factors must be considered, including the structure, physical properties, microfibrillar angle, defects, chemical properties, cell dimension, and the interaction of the fiber with the matrix [11,47,48,49,50,51], impurities [52], moisture absorption [53], orientation [54], volume fraction [55] and physical properties [56], which are inherent fiber characteristics, play an essential role in determining the mechanical properties of natural fiber-reinforced polymer composites (NFRPC). The mechanical properties of NFs are listed in Table 2.

A low density, good mechanical properties, recyclable quality, and high strength are all characteristics of these fibers that make them ideal for use as reinforcement in FRPC. It is widely recognized that the use of NFs, particularly as reinforcement in composite materials, is a preferred technology for a variety of sustainable industrial applications [68]. Naturally derived fiber-based composite materials have a wide range of applications in the industrial components, automotive industry, building structures, furniture, and packaging. They contribute to environmental sustainability by producing sustainable materials as alternatives to synthetic or traditionally made fibers [69]. NFRPC has numerous appealing advantages for lowering material costs and weight and providing sustainable solutions and is especially useful in plastics, electronics [70], packaging [71], and the automotive industry [72,73,74]. NFRPC is used in consumer applications such as interior paneling, domestic tables, window panels, and chairs [25,47,75]. These materials can also be used as an environmentally friendly alternative in automobiles and aircraft interior paneling. 

### 2.1. Benefits of NF over Synthetic Fiber Used in Sustainable Industrial Applications

Historically, the manufacturing of composite materials based on reinforcing two or more fiber types in a matrix has played a dominant role in a wide range of applications. Environmental concerns resulted in the development, use, and eventual removal of conventional fiber-reinforced synthetics. Because of this, the growth of natural fiber composites is accelerating as the need for green technology and renewable composites continues to expand. Due to the demand for environmentally friendly materials, synthetic material has shifted to natural fiber composites, to which more attention is being paid [76]. It is encouraged that more research is conducted on this subject, as NFs are derived from renewable plant resources that are easily biodegradable.

Siregar et al. [27] investigated the mechanical characteristics of epoxy composites reinforced with hybrid sugar palm/ramie fibers. Since ramie fiber offers some environmental advantages over synthetic fibers, ramie is an eco-friendly fiber resource. As an example, it provides health benefits while also releasing less carbon throughout the manufacturing process. Srinivasan et al. [77] examined the mechanical characteristics of a composite made of banana fibers and epoxy (particulate) reinforcement. Although synthetic fiber provides the greatest strength in a composite material, it has an issue with recyclability. Because of these aggravating issues, NFs have frequently been suggested as a suitable replacement for synthetic fibers to overcome problems. 

Moustafa et al. [78] conducted a review on environmentally friendly polymer composites used in green packaging materials. Because of their abundance, low cost, and environmental friendliness, lignocellulosic fibers are among the most useful NFs for reinforcing materials. Similarly, Rana et al. [79] discussed the effects of various surface functionalization approaches on mechanical, thermal, chemical resistance, water absorption, and moisture absorption properties of *Grewia optiva* fiber. NFs, which are readily available, environmentally friendly, and have a low density, are fast emerging as viable reinforcement materials in the composites sector and for the purification of polluted water.

Boland et al. [80] investigated the life cycle energy consumption and greenhouse gas (GHG) emissions of a 3 kg 30% glass–fiber composite component in automotive applications against a 30% cellulose–fiber composite component and a 40% kenaf–fiber composite component. It was found that the environmental benefits of NFRPC are primarily due to the lighter weight than glass–fiber composites, which increases the energy efficiency of automobiles. There are three main approaches for end-of-life disposal that have been considered: landfilling, incineration with energy recovery, and recycling. A study performed by [81] discovered that the amount of energy recovered from the burning of NFRPC ranged from 12 to 34 MJ/kg of the composite, which is greater than the amount of energy recovered from the incineration of glass fiber composites, which is around 7.5 MJ/kg.

### 2.2. Fabrication and Technique Process of NFRPC

Processing conditions, such as temperature, curing time, pressure, and manufacturing techniques used to develop the composite material, are critical in achieving an NFRPC with good mechanical properties. If any foreign particles enter the composite material during the manufacturing process, a composite with inferior properties is produced. Accordingly, the processing technique used to manufacture significantly impacts both the desired characteristics of the composite material and the characteristics of the part manufactured from the composite material. Studies of processing technologies such as hand layup, resin transfer molding, hot press, compression molding, injection molding, and extrusion are critical. 

Khalid et al. [82] evaluated the tensile strength of glass/jute fibers reinforced composites. The hand layup technique was chosen because of its numerous advantages, including its ease of use and low cost during the manufacturing process. Mold-releasing wax was applied to the glass mold surface, then oriented the peel-ply and various fibers one by one in the desired stacking sequences. Excess epoxy from roller movement leaks out of the specimen sides and typically adheres to the glass mold surface. Figure 4 depicts the necessary tool and material arrangements for the hand layup technique.

Sarikaya et al. [83] developed an epoxy resin composite reinforced with birch, palm, and eucalyptus fibers using a combination of resin transfer molding (RTM) and molded fiber production techniques to produce the composites. A vacuum-assisted RTM system was developed to inject resin into molded fiber plates and cure the resin. The resin and hardener were mixed in the preferred ratios of 100/34 (*w/w*). RTM system was used to place the dry fiber plates between airtight molds. The resin/hardener mixture was injected into the molds from one side. The other side of the plate was also vacuumed simultaneously to achieve a homogeneous distribution of the resin/hardener mixture without the presence of air bubbles. The mixture ratio of resin and hardener/fiber is applied as 13/7 (*w/w*), respectively. Following the injection process, the molds were heated up by cartridge heaters to complete the curing process. Curing took 2 h at 100 °C. Figure 5 shows resin injection into the molded fiber plate.

Obada et al. [84] investigated the physical, mechanical, and dynamic mechanical properties of coir (coconut fiber) and coconut husk particulates reinforced polymer composites prepared by the hot press method. The composition of coconut husk powder, coir, and shredded plastic water sachet (RLDPE) was weighted and mixed for 30 min at a temperature of 180 °C using a two-roll mill. In order to thermoform the composites, a steel mold was used in conjunction with a hot press machine. The working temperature was 80 °C, with a 15-min preheating period, followed by 2 min of applied compaction pressure and 5 min of cooling. 

Another study performed by Zhao et al. [85] was related to the preparation of polypropylene (PP) film, which was used as a carrier to wrap long bamboo fibers (about 120 mm long), and the PP roll that contained the bamboo fibers was fed into and compounded by a twin rotor. The extruded mixture was then hot-pressed into plates. The compounding process was monitored and analyzed. The fiber size measurements revealed that a rotor with a smooth arris and a lower arris number reduced the damage to LBF. When the LBF content was set to 40%, the optimal LBF distribution and orientation in the PP matrix were obtained.

Furthermore, a study conducted by Chen et al. [86] focused on the preparation of high-grade textile materials and filtration membrane materials using bamboo cellulose. Bamboo cellulose functionalized modified material by the ionic liquid. The separation technologies of bamboo cellulose were reviewed from the standpoint of bamboo cellulose utilization, including pre-hydrolysis technology, cooking technology, bleaching technology, selective removal of hemicellulose from bleached chemical bamboo pulp, and upgrading technology to prepare to dissolve pulp. Fiber morphology, which includes fiber length (maximum, minimum, and average), cell wall thickness, and its derivatives, was investigated. Chemical analysis of bamboo samples was also conducted. This renewable biomass, moso bamboo (Phyllostachys edulis), has a high degree of mechanical strength, anisotropy, and fine multi-scale hierarchical structure. A similar study on the application of ionic liquids to treat natural fibers was conducted by Chunhui et al. [87].

Anilkumar et al. [88] investigated the mechanical properties of natural fiber–polymer composite materials. Compression molding was used to create the coir–vinyl ester composites incorporated with filler. The rice husk and eggshell particulates are combined with polyester resin through rapid mechanical shaking. The resulting mixture is poured into molds in the compression molding system, where suitable pressure is applied. Figure 6 shows the schematic diagram for compression molding machines.

Sathees et al. [89] studied the effects of fiber loading on the mechanical characterization of pineapple leaf and sisal fibers reinforced polyester composites. The injection molding method was used to create composite samples with varying fiber weights. The NFs and polyester resin were mixed before being injected into the injection molding machine to produce the final product. The blended materials flowed through the molding chamber and were poured in a rectangular formed shape (160 mm × 140 mm × 4 mm) for 10 min at a tension of 10 MPa and a temperature of 190 °C. During the cooling stage, a similar weight was held in place for 15 min. The fabrication process of the natural fiber composite specimen is shown in Figure 7.

A study conducted by Hao et al. [90] looked into the effects of fiber geometry and orientation distribution on the anisotropy of the mechanical properties, creep behavior, and thermal expansion of natural fiber/high-density polyethylene (HDPE) composites. The specimens were fabricated using the extrusion method. A specific ratio (60:35:3:2) of dry NFs, HDPE, Maleic anhydride grafted polyethylene (MAPE), and lubricant were mixed for 10 min at 80 °C in a high-speed mixer (1500 r min1). They were melted and blended in a co-rotating twin-screw extruder and then extruded through a single-screw extruder to fabricate a profile with dimensions of 100 mm (width) and 4 mm (thickness). Flexural, impact, creep, and thermal expansion tests were performed on specimens cut from extruded profiles at various inclination angles (0°, 30°, 45°, 60°, 90°) to the extrusion direction Figure 8. 

A study was conducted by Azmin et al. [30] to create biodegradable plastic sheets using sugarcane bagasse and cocoa pod husk through the solution casting method. In a 250 mL beaker, all of the ingredients were prepared and blended, including 1.5 g bagasse fibers, 1 mL glycerine, 40 mL distilled water, and 0.5 g sorbitol. The solution was then stirred on the hot plate for 30 min until evaporation occurred and the solution became viscous. The mixture was then poured and spread in a glass petri dish to create a fiber-based plastic. The bioplastic film was oven dried for an hour at 50 °C before being dried at room temperature for two days. Sugarcane bagasse and cocoa pod husk were used to extract cellulose and fiber, respectively. The created bioplastic sheets were separated into numerous cellulose and fiber concentration ratios, including 100:0 (100% cellulose), 75:25 (cellulose:fibre), 50:50 (cellulose:fibre), 25:75 (cellulose:fibre), and 0:100 (100% fiber). All bioplastic concentration ratios’ physical and chemical characteristics were assessed in terms of sensory assessment, drying time, moisture content, water absorption, and water vapor permeability. The combination of 75% cellulose and 25% fiber bioplastic, which showed the lowest water absorption percentage, was deemed to be the best acceptable bioplastic film for food packaging, according to observation and study of the physicochemical features of bioplastic.

Oyeoka et al. [91] developed a study related to the packaging and degradability properties of polyvinyl alcohol/gelatin nanocomposite films filled with water hyacinth cellulose nanocrystals. The films were made using a solution casting method. A predetermined weight of PVA (as determined by the experiment design) was dissolved in distilled water at 90 °C for 30 min with constant stirring until complete dissolution was obtained, yielding a 100 g solution. A known weight of gelatin (as specified in the design) was also dissolved in separate distilled water at 60 °C. The PVA and gelatin were then mixed together for 30 min at 90 °C with continuous stirring. At room temperature, the required amount of water hyacinth CNC was dispersed in glycerol (30% of the total solid content of gelatin and PVA) and added in drops to the PVA/Gelatin solution while stirring at 90 °C. After 30 min of stirring, the mixture was poured into a 130 mm diameter petri dish and air dried for 48 h to ensure slow evaporation of the solvent. The resulting films were peeled from the dish and placed in a desiccator until they were evaluated. With the addition of CNCs, the thermal stability of the films was increased from 380 °C to 385 °C. The addition of CNC to the PVA-gelatin blends reduced water absorption, water vapor permeability (WVP), and moisture uptake of the films. Moisture uptake decreased from 22.50% to 19.05%, with the least WVP being 1.64 × 10^−6^ g/(m·h·Pa) when 10% CNC was added. These findings suggest that WHF CNC and PVA-gelatin blends could be used in biodegradable films for on-the-go food wrappers.

## 3. Modifications Treatment on NFRPC 

### 3.1. Physical Treatment—Corona, Plasma, Superheated Steam

Figure 9 shows a summary of the physical treatment characteristics of natural fiber for reinforced polymeric composites. In general, the function of physical treatment depends on the structure of reinforced polymeric composites that needs to be produced. There are two main functions of physical treatment, namely, separate natural fiber bundles into individual filaments and modification of the fibers for composite applications. For the first function, there are two phases of the method that were used, namely, the former and the current. The former methods include chemo-mechanical, water-retting, and dew-retting. This method involves the chemo-mechanical treatment, which tends to produce unacceptable fermentation waste [92]. Therefore, physical separation processes were adapted, such as the steam explosion process and thermomechanical process, for the same functional capacity. Both processes separate the lignocellulosic material into its main components, namely cellulose fibers, amorphous lignin, and hemicellulose. A single fiber postulated has greater strength and stiffness than the fiber bundles, leading to increasing hydrophobicity of the surface and improving the capability with the matrix such as polypropylene [92].

Moreover, for the second function of physical treatment, the modification of the fibers for composite applications can be divided into two phases, namely thermal and non-thermal. For thermal, the method is known as plasma treatment. Plasma treatment is a method to bring a physical modification to the surface through roughening of the fiber by the sputtering effect, producing an enlargement of the contact area that increases the friction between the fiber and the polymer. It is divided into two types, namely vacuum and atmospheric, which, for vacuum type, requires the parts to be treated under low vacuum pressure in the chamber. Meanwhile, atmospheric plasma pretreatment techniques are highly attractive since the parts are treated in situ rather than in a chamber. Both vacuum and atmospheric produce better adhesion characteristics and promote the action of coupling agents [92].

In non-thermal treatment, it promotes the polymer surface to become more wettability and adhesion. The two methods for producing non-thermal are namely corona discharge and dielectric-barrier discharge (DBD). Corona is emphasized as one of two aspects of the discharge; the energetic electron produces the plasma or the ions, while the dielectric barrier concentrates on generating the ozone. The most typical corona configuration (continuous and pulsed) is created around a sharp edge, which leads to maximizing the active discharge volume. Continuous corona discharges are limited by low current and power, which results in more applications for materials and gas streams (environmental and fuel conversion applications included). It is possible to increase power in a corona discharge (without transition to the spark regime) by using pulse-periodic voltage. Therefore, pulsed corona can be relatively powerful (10 kW) and quite luminous. In the aspect of trying to find a solution for avoiding arc formation, dielectric barrier discharge is similar to pulsed corona. However, DBD does not require such complicated pulse power supplies. Due to the simplicity of its operation in strongly non-equilibrium conditions at atmospheric pressure and at reasonably high-power levels without using sophisticated pulse power supplies, it became one of the important techniques; corona even had wide application in a variety of processes [92].

### 3.2. Chemical Treatment 

Table 3 shows the types of chemical treatments available to improve the characteristics of natural fiber. There are 12 types of chemical treatments that are listed, complete with chemical reagents used, methods, structure improvement, and application based on the natural fiber used. Among the chemical treatments described in Table 3 are alkaline, saline, acetylation, alkaline hydrogen peroxide, benzoylation, acrylation and acrylonitrile grafting, maleated gents, permanganate treatment, peroxide treatment isocyanate treatment, ionic liquid, and thermal decomposition kinetics.

### 3.3. Mechanism of Fiber Treatment 

The natural fibers with limited thermal stability, which are normally up to 230 °C usable temperature and exhibit high moisture absorption due to the presence of hydroxyl and other polar groups, are a major drawback associated with the application of biofibers for reinforcement of polymeric matrices [92]. Accordingly, physical and chemical treatments were adapted in the production of NFRC to expand the use of biofibers in the industry. Physical treatments that are always used for surface modification and increased mechanical bonding between the fiber and the matrix are corona, plasma, mercerisation, and heat treatment [93]. Physical treatment does not extensively alter the chemical composition of the fibers and clean them since they do not involve chemicals. The mechanism of each physical treatment is shown in Table 4. Among the physical treatments listed in Table 4, corona treatment is the best because it has many advantages compared to others on surface modification and also easily modified the instrumentation used led to continuous process in industry.

In addition to physical treatment, chemical treatment is also used in the modification of fiber surfaces. Chemical methods can be divided into four major categories: esterification-based treatments, silane coupling agents, graft copolymerization, and treatments with various chemicals. Dimethylurea (DMU) and phenol formaldehyde (PF), listed under various chemical categories in Table 4, are the most notable but rarely used in natural-fiber composites. Among the chemical treatments listed in Table 4, graft copolymerization with maleic anhydride is the best available method for natural-fiber composites as it consistently delivers composites with superior properties. Details related to the mechanism of the chemical treatment and the properties improved after treatment are shown in Table 4.

## 4. NFRPC for Sustainable Industrial Applications

### 4.1. Potential of NFRPCs in Industrial Applications

The versatile characteristics of NFRPCs make them ideal to be used for numerous applications ranging from household goods to more advanced and specific end-use. In fact, NFRPCs are already marketed for different applications, including in packaging, automotive, medical products, aerospace, sports, and others. This section focuses on the use of NFRPC in various industrial applications.

#### Automotive

In the automotive field, natural fibers were used as reinforcing materials with different polymer matrices. The earliest reported application of NFRPCs was back in the 1940s, when Henry Ford used hemp fiber composite to produce components in a car. Meanwhile, in 1996, Mercedes-Benz prepared door panels by using jute fiber as a replacement for synthetic fiber [109]. Following that, there was a rapid growth in the use of NFRPCs in the automotive industry prompted by the intensive research conducted, environmental concerns, and the new policy and vehicle legislation. This applies to the legislation by the European Union End of Life Vehicle Directive (2020), in which 80% of a vehicle must be recyclable, recovered, or reused. Currently, NFRPCs are commonly used for interior and exterior vehicle components such as dashboards, headliners, door panels, seat backs, decking, noise insulation panels, boot lining, hat racks, etc. NFRPCs are now being preferred in the automotive industry compared to their synthetic counterparts as the use of natural fibers in composites is able to meet the performance required in terms of strength and durability [110]. More importantly, they are also lightweight, which can reduce 25% of vehicle weight [111], have lower production costs, lower energy consumption, and are recyclable. In this perspective, NFRPCs possess both technological and environmental benefits. Figure 10 exhibits some examples of commercialized car components made up of natural fiber-reinforced composites.

### 4.2. Building and Furniture

Other than automotive, the applications of natural fiber reinforced polymer composites are also found in the building and furniture industries. NFRPCs can be used for the manufacturing of door frames, windows, floor matting, partitions, and ceilings. Additionally, they can be used to make tables, chairs, and other kitchen tools [113]. NFRPCs are gaining interest in this sector because they can create products with both required properties and aesthetic value. Natural fibers such as hemp, jute, bamboo, and other types were used for this purpose. Figure 11a shows the natural fiber biocomposites chair made up of hemp with epoxy resin [114]. Moreover, Figure 11b,c display fully biodegradable composites of PLA–Bamboo in cylindrical concave and concave–convex shapes. The composite demonstrated excellent mechanical behavior as well as the ability to produce structures with complex geometries [115].

### 4.3. Military

One example of the utilization of NFRPCs in the military field is for the production of personal body armor. Generally, body armor is designed to absorb or deflect physical attacks. They can be classified into soft body armor and hard body armor or multilayer armor system. They differ based on their composition and their protective capability. Soft body armor is made up of multiple layers of fabrics up to 50 layers and weighs below 4.5 kg. It can withstand up to 500 m/s impacts of projectiles. Meanwhile, hard body armor refers to the combination of soft armor and hard plates. It can withstand an impact of more than 500 m/s based on the National Institute of Justice (NIJ) armor standard [116]. Depending on the design, the panel used in the hard body armor can be made from different compositions, such as ceramics/composites, ceramics/metal, composites/metal, and ceramics/composites/metal. The most commonly used combination in defense is the ceramics/composites/metal combination. Materials such as aramid fabrics (Kevlar and Twaron) and ultrahigh molecular weight polyethylene (Dyeenama and Spectra) are the most popular as a second layer for multilayer armor [117]. Despite that, these materials are costly and obtained from non-renewable resources. Currently, NFRPCs have become a hot topic of investigation because of their performance compared to conventional aramid fabrics [118]. From this point of view, various types of natural fiber were investigated. 

In one study by Santos and co-workers [119], pineapple leaf fiber (PALF)/epoxy composites were used as the second layer protection. The front layer was made of a ceramic layer, and the two layers were joined together by polyurethane adhesive (PU). The sample was prepared by a compression molding technique. The PALF used was continuous fiber and was not treated to minimize cost and reduced the production stage. The properties of the samples were investigated with ballistic tests and a back-face signature (BFS) test. In this study, single ceramic layer was used as a reference. They found that the ceramic single layer recorded a BFS reading of 35.9 mm, whereas the ceramic/PALF composites exhibit an improved ballistics performance with a BFS depth of 26.6 mm, which is almost the same value as Kevlar material. Based on the NIJ standard, the acceptable value of BFS must be lower than 44 mm for the materials to be considered effective protective armor. Figure 12 shows the illustration of the hard armor system with NFRPC for bulletproof vests, as proposed in their study [119].

Another study on woven kenaf/Kevlar hybrid yarn for anti-ballistic composite material was investigated by the Jambari team. They study the effect of hybridization on the properties of the kenaf/Kevlar/epoxy composites. The samples were prepared by hand lay-up method with 40% fiber content and 60% epoxy as the matrix. The kenaf/Kevlar ratios were varied in weight fraction of 30/70, 50/50, and 70/30. Kenaf/epoxy and Kevlar/epoxy composites were used as a comparison. They found that the Kevlar/epoxy showed superior performance compared to the kenaf/epoxy composite, which was expected due to the poor mechanical properties of kenaf. Hybridization had intermediate effects on mechanical properties. Based on the ballistic testing result for hybrid composites, yarn kenaf/Kevlar with a 30/70 ratio showed the highest ballistic limits with maximum energy absorption of 148.8 J at 318 ms^−1^ striking velocities with a target thickness of 12 layers [120]. In addition, a study on Ramie/Kevlar/polyester composites for solid body armor was conducted by Radif and colleagues [121]. The Ramie/Kevlar layers were impregnated using polyester resins to create an interfacial linkage between the layers. This study revealed that the arrangement of the Ramie/Kevlar layers and their thickness played an important on the ballistic performance. Moreover, their Ramie/Kevlar composites also met the third level of the NIJ standard by increasing the number of panels [121]. Furthermore, a study on the ballistic performance of hybrid Kevlar/coconut sheath (CS) reinforced epoxy composites were reported previously. In this study, hybrid and non-hybrid laminates were prepared with different layering sequences by hand lay-up and hot press method. The hybrid composites exhibited higher ballistic limits compared to the Kevlar/epoxy composites due to the areal density differences [116].

### 4.4. Packaging

The majority of packaging materials currently used are served by a non-biodegradable polymer that has a long-life span and is nearly indestructible. The persistence of these materials in the environment beyond their functional life has resulted in a broad range of pollution, litter, and waste disposal problems [122]. Concerns about the growing environmental issues and preservation of natural resources have triggered researchers and industries to develop products and processes compatible with the environment. The use of natural fiber reinforced polymer composites sparked a great interest in recent years as a way to alleviate the problem related to conventional products [123]. As a packaging material, NFRPCs are readily extruded and could be manufactured for different types of products with necessary demanded properties. An investigation on the life-cycle environmental impacts of a fully bio-based sugar palm fiber (SPF) reinforced sago starch for takeout food containers was reported previously. The actual concept design of the container is displayed in Figure 13. This study conveyed the life-cycle assessment (LCA), especially on the damage assessment of the whole process, from the production stages to the disposal of the SPF food container. This is conducted on the basis of two important categories, which are human health and ecosystem damage. In the case of human health, disability-adjusted life years (DALY) were used as the indicator. DALY refers to the difference between the ideal situation of the standard life expectancy in perfect health and the actual situation. The unit is measured from 0 (safe/no effect), 0.67 and above (danger/critical) and 1 (fatal/death). Moreover, the ecosystems were expressed as the loss of species over a particular area during a specific time. For the assessment of the human health damage category, the result showed below 0.0001 DALY, whereas below 0.00001 species.yr was recorded for the ecosystem damage category. These results indicate that the bio-based food container could lower environmental and human health damage [124].

Lai et al. reported the use of microcrystalline cellulose (MCC) derived from oil palm empty fruit bunch fiber (OPEFB) and nano-bentonite (NB) to prepare thermoplastic starch (TPS) hybrid composites for packaging material. The MCC was obtained by the chemical treatment and acid hydrolysis of OPEFB. From this study, the addition of MCC/NB into the TPS able to produce a composite with improved strength, modulus, toughness, and flexibility, which are useful criteria for the production of flexible packaging film [125]. In one study, composite polymer films from poly(3-hydroxybutyrate-co-3-hydroxyvalerate) (PHBV) and *Ceiba pentandra* (CP) bark fiber were prepared as a packaging material for strawberries. From this study, it was found that the addition of CP bark fiber enhanced the tensile properties, crystallinity, thermal properties, and percentage of biodegradability. Interestingly, at high fiber loading, the composite films showed antibacterial activity against *S. Aurus*. This is due to the presence of lignin in the fibers, which contain a phenolic compound that inhibit the growth of certain microbial organisms. The finding also indicates that, after 7 days, strawberries packed with films containing more than 10% CP fiber have better preservation of freshness of the fruit. This shows that the addition of natural fiber can be used as an eco-friendly active packaging to extend the shelf life of perishable fruits [126]. 

### 4.5. Sport Equipment

Moving forward, another important application of NFRPCs is for the production of sports equipment. Traditionally, materials such as steel, aluminum, alloy, and wood were used to produce sports products. It is a fact that, as most sports require movement, the sports equipment is preferably lightweight. Therefore, currently, natural fiber-reinforced composites based on lightweight products are being used in sports equipment. Research and development in this area are continuously evolving to find materials with improved strength and functionalities [127]. To date, hybrid fiber–polymer composites with natural and synthetic fiber have been used to produce sports equipment. One example of fiber that has been used for commercial sports products is flax. For instance, flax/carbon fiber (25:75%) reinforced tennis rackets, flax/carbon fiber (80:20%) reinforced bicycle frames, and flax reinforced snowboards. It is claimed that flax fiber provides superior dampening properties compared to carbon and glass fiber and thus improves product quality [128]. Other applications include fishing rods, archery, and ski poles [129]. In one study reported by Yusup et al., oil palm empty fruit bunch (OPEFB) fiber/epoxy composite was prepared, and their characteristics and potential for field hockey sticks were investigated. In this study, they examined the suitable duration for the alkaline treatment of the OPEFB fiber at 12 h and 24 h. Based on the mechanical properties result, treatment of the OPEFB fiber at 24 h showed high potential to be used as the reinforcement to epoxy for the hockey stick purpose [130].

### 4.6. Medical Application

The application of NFRPCs in the biomedical field is not a new concept. In fact, they have been used for drug delivery, tissue engineering, orthopedics, medical implants, wound dressing, cosmetic orthodontics, and others. The production of NFRPCs for medical uses depends on the intended applications. One of the important criteria to be considered for the NFRPCs to be utilized in the medical field is their acceptability and compatibility with the human body [131]. In one study by Mangat et al. [132], they prepared natural fiber inserted three-dimensional structural composites fabricated with fused filament deposition by using a low-cost destock printer for scaffold-based biomedical application. In this case, silk fiber and sheep wool fiber were used as laminations and polylactic acid as the polymer matrix. Prior to use, the fibers were subjected to chemical treatment to remove impurities. The mechanical properties and antibacterial properties of the composites were investigated. It was observed that the composites showed antibacterial properties against *E. coli* and *S. aureus*. Additionally, it was found that the type of fibers determined the dimensional accuracy of the structures, while the density and number of laminate affected the flexural strength and hardness of the specimen [132]. Another example of the utilization of natural fiber for biomedical application was reported by Rahman et al. In this work, nano and microcrystalline cellulose (CC) was extracted from jute fiber by the optimization of different reaction parameters to obtain a high extraction yield. The composite films were prepared by mixing CC (3–15%) with polylactic acid. Based on the antimicrobial and cytotoxicity study, PLA with 15% CC showed the best antimicrobial effects against *E. coli* and *B. subtilis*. In addition, the samples also showed non-toxic properties. The PLA/CC film has the potential to be used as parent material for bone implants [133].

### 4.7. Cost Effectiveness of NFRPCs in Sustainable Industrial Applications

Natural fibers are known to offer several environmental and economic benefits as they are readily available, abundant in nature, recyclable, biodegradable, lightweight, and, most importantly, they are relatively inexpensive compared to synthetic fibers [69,134]. In view of this, composites made up of natural fibers and polymer matrices are of major interest because they can provide desirable properties at a low cost. Though there are some challenges in regard to the utilization of natural fibers in terms of compatibility with current processes, a good balance between composite performance and production cost could be achieved with the proper selection of materials and designs [135]. The cost-effectiveness of using these materials is one of the main attractions and motivations for using NFRPCs in industrial applications. In some light, the development of NFRPCs can fit well with the sustainability concept of producing sustainable products. The concept of sustainability is based on three main cores, which consist of the environmental, social, and economic points of view, as illustrated in Figure 14 [123]. In essence, the production of NFRPCs consumes lower energy than conventional synthetic fiber reinforced composites, have a low environmental impact, is eco-friendly, suitable to be used in various application, and cost-efficient [133]. In order to achieve the sustainable goal, it is necessary to follow the standards of consumption and production, from the selection of raw materials, technology used, and the production process to the final products.

The most common natural fibers used for NFRPCs are flax, hemp, kenaf, cotton, wood, etc. Among them, wood fiber dominated the market with 59.3% of the total revenue in 2015 and is expected to continue its dominance until 2024 [135]. Additionally, flax was one of the most widely used fibers after wood, with a market share of 13%. In 2016, the global market size of natural fiber composites was valued at USD 4.46 billion. It is reported that NFRPCs are widely used in the construction and automotive field, followed by electronics and sporting goods [136]. Currently, the top manufacturers of the NFRPCs are led by Trex Company, Inc., Winchester, VA, USA, The AZEK Company, Chicago, IL, USA, Fiberon LLC, New London, CT, USA, Avient Corporation (PolyOne), Shirur, Maharashtra, India, Oldcastle Architectural (AERT), Austin, TX, USA, Anhui Sentai WPC Group, Xuancheng, Anhui, China, UPM Biocomposites, Bruchsal, Germany, Tecnaro GmbH, Ilsfeld, Germany and others. By far, Trex Company, Inc. has the largest market share, accounting for more than 10% of the total market, while the Asia-Pacific region is the largest market for NFRPCs, with more than 30%. The forecast for NFRPCs is expected to have a compound annual growth rate (CAGR) of 9.4% in 2021–2027 [137]. Even though the market shows continued demand and growth for NFRPCs over the year, there are some valid concerns about the utilization of these materials. Firstly, issues with the compatibility of the natural fibers with the polymer matrix. The natural fibers are hydrophilic in nature, which makes them incompatible with the hydrophobic polymers. Due to this, extensive research is being conducted to tackle this limitation. In addition, the complexity of the process somehow restricts NFRPCs application. Another important factor is on the geographic barrier as some of the manufacturers cannot easily access to the natural fiber. The origin of the natural fiber also produces variations in properties, stability, and durability of the composites. Moreover, the user has not been well informed on the benefits of using NFRPCs, which limits their application [138]. In view of this, Europe is predicted to remain as the largest market for the NFRPCs in coming years due to their policy and high acceptance level on eco-friendly composites by the industries and government [69].

### 4.8. A Roadmap of NFRPC to Moving into Industry 4.0

NFRPC has become a significant domain of research due to the availability of a wide range of plant fibers as reinforcements that are both easily available and biodegradable [139]. There was numerous research conducted on the use of natural fibers as polymer composite reinforcements, and they have grown in importance for a variety of industrial applications. According to Rohit and Dixit [140], the most recent development of NFRPC is for: (i) transportation industries such as automobiles, railway coaches, and aerospace; (ii) building and construction industries such as ceiling paneling and partition boards. It is because NFRPC has qualities that are comparable to, if not better than, traditional polymer composites [141]. This section presents the growth of global interest in NFRPC application in various industrial applications and highlights some of the research associated with the NFRPC to achieve long-term sustainability. 

The use of natural fiber has undergone numerous revolutions over the years, from synthetic polymers to natural fibers [142]. The research on NFRPC is continuously developing and has contributed to the increased ability of natural fibers applications in various industries. A total of 394 documents were obtained from the Scopus database based on the following search strategy criteria (TITLE-ABS (“Natural Fiber Reinforced Polymer Composites” OR “NFRPC”)) AND PUBYEAR < 2021 OR PUBDATETXT ((“January 2021” OR “February 2021” OR “March 2021” OR “April 2021” OR “May 2021” OR “June 2021” OR “July 2021” OR “August 2021” OR “September 2021” OR “October 2021”)) AND (EXCLUDE (PUBYEAR, 2022)) AND (LIMIT-TO (LANGUAGE, “English”)).

For 17 years, since 2005, 20 documents have been published in the NFRPC research area for various industrial applications. Fifty percent of the total documents were published as journal articles, while 20% were published as conference papers, highlighting a great demand for state-of-the-art studies to further this new line of research. Thirty percent of the total documents were, respectively, published as review papers and book chapters. Commonly, the applications of NFRPC is suggested for automotive [47,143,144,145,146], furniture [142], textile [147], building components [148], and other various industrial applications (see Table 5).

The important breakthroughs of NFRPC applications are associated with the demand of industry for sustainable materials. According to Alkbir et al. [145], NFRPC can be used in an extensive range of applications due to their fairly good mechanical properties, low cost, high specific strength, environmental friendliness and bio-degradability, ease of fabrication, and good structural rigidity. 

It is also fascinating to see how NFRPC and IR4.0 are linked. There were 394 articles written for NFRPC in total (the minimum number of keyword co-occurrences is four). Only 59 keywords out of 914 met the threshold. After refining the process by removing the duplicate keywords, 40 met the criteria. Despite the fact that 12,819 articles were created for the IR4.0 scope, there is no network of co-occurrences for the combination of NFRPC and IR4.0. Based on the density visualization (Figure 15), the majority of NFRPC research articles focused on mechanical characteristics, natural fiber, composite materials, and polymer composites. Currently, there are no industrial studies on the production of the NFRPC. As a result, researchers must fill in the gaps and conduct future studies.

The data were then examined for link strength. The network visualization shown in Figure 16 was used to establish how strong the links of research on NFRPC with any nine major IR4.0 technologies are, such as Big Data Analytics (BDA), Optimization/Simulation, Cloud technology, Virtual/Augmented Reality (VR/AR), Horizontal/Vertical System Integration, Industrial Internet of Things (IIoT), Additive Manufacturing (AM), Autonomous Robots, and Cybersecurity. 

Only 16 of the 104 keywords passed the criterion after further refining, which included deleting duplicate phrases and lowering the minimum number of keyword co-occurrences to two. Surprisingly, the NFRPC network diagram shows that the study’s primary focus is on sustainability rather than IR4.0. Under the sustainability term, there are three links: natural fiber, polymer composites, and chemical treatment (four total link strengths with two occurrences).

## 5. Challenges and Future Perspectives

It is evident from the review above that NFRPC use in sustainable industrial applications is justified due to their mechanical strength being comparable to synthetic fibers and having less of an impact on the environment. However, enhancing and managing the mechanical characteristics of NFRPCs is extremely difficult. In order to support and promote the utilization of novel NFs as well as novel chemical techniques in the advancement of NFRPCs, further investigation is also required from the research community. 

There is a knowledge gap regarding the safety, durability, and, most importantly, recyclability of NFRPCs. When recycling composites that contain formaldehyde-based adhesives that emit volatile organic compounds, there may be safety risks. If the composites are used in exterior applications, where they are not protected from weather, biological attack, etc., durability is an issue. Natural fiber composites are recyclable by definition. Polymer and fiber degradation, high moisture content, flammability, variation in natural fiber composition, and poor bonding between hydrophilic fibers and hydrophobic polymers are the major technical challenges for the recycling and application of NFRPCs that must be addressed. Understanding how to improve the properties of NFRPCs before and after recycling is critical for their increased use.

We may conclude that NFRPC development is accelerating and is envisioned as a future sustainable material for new applications. However, future research is required to overcome the challenges in developing NFRPC for sustainable industrial applications, such as machine and material cost reductions, as well as limited technological adaptation. Furthermore, NFRPC has low mechanical properties due to fiber-matrix incompatibility and natural fibers’ inherently weaker nature when compared to synthetic fibers. Thus, one way to address the shortcomings in the way natural fibers interact with the polymer in NFRPCs is to modify the natural fibers. Alkalisation, for example, alters the composition of the fiber structure and significantly reduces the natural fibers’ capacity to absorb moisture. This results in improved interfacial adhesion between the fiber and the polymer matrix.

Despite the limitations, the use of NFRPCs is currently expanding positively, particularly in the automobile industry. Hemp, kenaf, flax, and bast fiber are now used in automotive components. In addition, the wood plastics in composites make them perfect for use in the construction industry. NFRPC are currently being used successfully in electrical devices and sporting goods; thus, they have a strong chance of capturing a significant market share.

The development of new materials, fiber-matrix interfacial characteristics, fiber homogeneity, fiber alignment, interlayer bonding, porosity, and printability are all future focus areas related to the NFRPC. The production and attributes of fiber for NFRPC can be impacted by changes in the quantity, price, quality, and features of the fiber supply during seasonal and non-seasonal times. Additionally, the hybridization of composite materials made from various natural fibers helps address a lack of fiber supply. The construction of processing and storage facilities close to agricultural land can result in significant cost savings for producing NFRPC.

In order to achieve long-term stability in outdoor applications, future research is required to tackle the challenges of NFRPC, including moisture absorption. Extreme weather conditions, such as temperature, humidity, and UV radiation, all have an impact on the service life of NFRPCs. In order to achieve high performance, emerging mitigation approaches during recycling can be used to achieve long-term durability and recyclability of composites.

## 6. Conclusions

This review study provided an in-depth look at the properties and benefits of natural fiber and NFRPC for sustainable industrial applications. NFRPC may be able to tackle the problems caused by non-renewable materials by using the circular economy notion to create environmentally friendly products and technology. With NFRPC’s long-term viability, a wide range of novel industrial applications can be fully realized. This review also encapsulated a broad range of NFRPC research on the processing techniques, modifications, treatment, applications, and the roadmap towards industry 4.0 for long-term industrial use.

We found that the following characteristics influenced the selection of sustainable composites due to the following qualities: (1) The properties of natural fiber are influenced by a variety of factors, including physical characteristics, chemical composition, crystalline cellulose dimensions, microfibrillar angle, defects, structure, and isolation technique. (2) The properties of NFRPC are influenced by the structure, impurities, moisture absorption, orientation, volume fraction, physical properties, microfibrillar angle, defects, chemical properties, cell dimension, and the interaction of the fiber with the matrix. Natural fibers were recommended as a possible replacement for synthetic fibers in order to lessen the negative environmental effects of modern materials. From the above review, it is clear that the benefits of natural fibers utilized in sustainable industrial applications were ecologically friendly materials, low cost, easily biodegradable, releasing less carbon (eco-friendly fiber resource), recyclability, abundance, and energy efficiency for end-of-life disposal.

The review also highlighted the broad range of methods used by researchers to fabricate and process the NFRPC, such as the hand layup method, combination of resin transfer molding (RTM) and molded fiber production techniques, hot press method, injection molding method, and extrusion method. Finally, the applications of NFRPCs in industrial applications were presented. Some of the examples of the utilization of NFRPCs are in the automotive industry (interior and exterior vehicle components such as dashboards, headliners, door panels, seat backs, decking, noise insulation panels, boot lining, hat rack), building and furniture industries (door frames, windows, floor matting, partitions, ceilings), military field (personal body armors, anti-ballistic composite material), packaging applications (food container, flexible packaging film), sports equipment (reinforced tennis racket, bicycle frames, snowboard, fishing rod, archery, ski poles), and biomedical field (scaffold-based biomedical application, parent materials for bone implants).

The primary concept of sustainability is based on three major cores that are interconnected: the environmental, social, and economic spheres of view. The industry’s need for environmentally friendly materials has resulted in major advancements in NFRPC applications. However, future work needs to be performed to investigate more potential applications of NFRPC. In the last 17 years, only 20 documents for various industrial applications have been published on the NFRPC study topic. Moreover, no article on the use of IR4.0 to speed up the production of NFRPCs in order to fulfill industry demand has yet been written. As a result, further study is needed in the future to overcome the barriers and lead to a circular economy that can address global issues such as climate change, biodiversity loss, waste, and pollution.

## Figures and Tables

**Figure 1 polymers-14-03698-f001:**
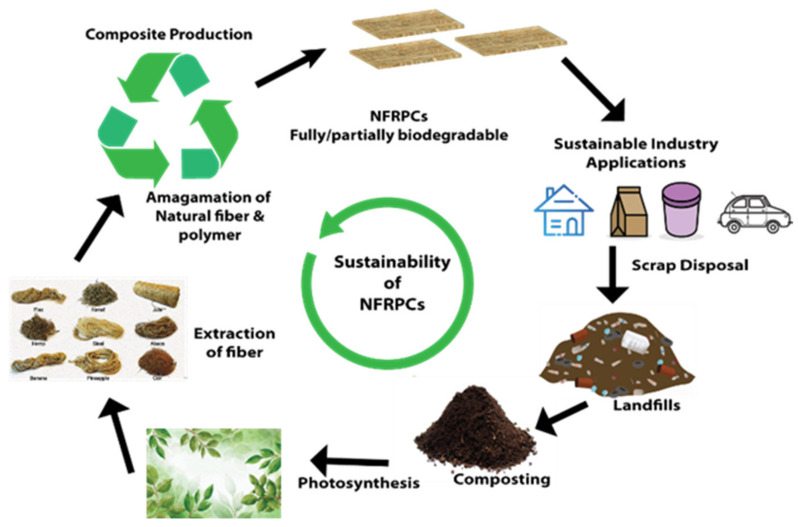
Sustainability of NFRPC.

**Figure 2 polymers-14-03698-f002:**
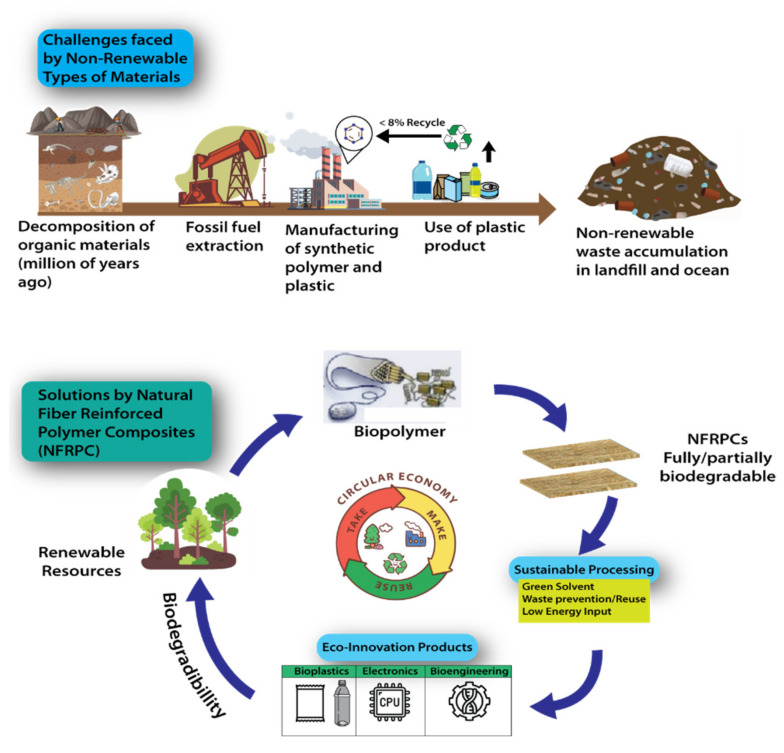
Challenges faced by non-renewable materials and solutions offered by NFRPC.

**Figure 3 polymers-14-03698-f003:**
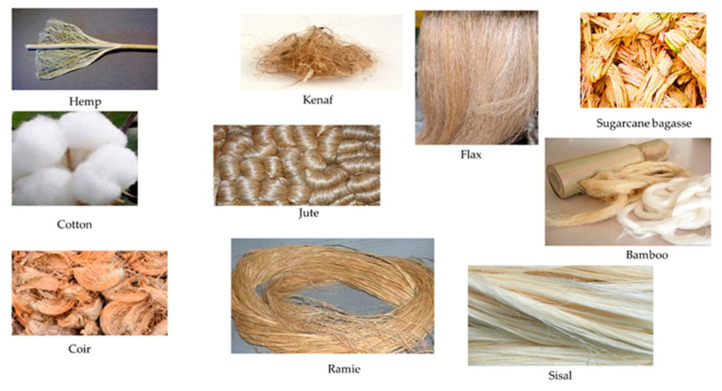
Different types of natural fibers. Reproduced with permission from Ref. [32].

**Figure 4 polymers-14-03698-f004:**
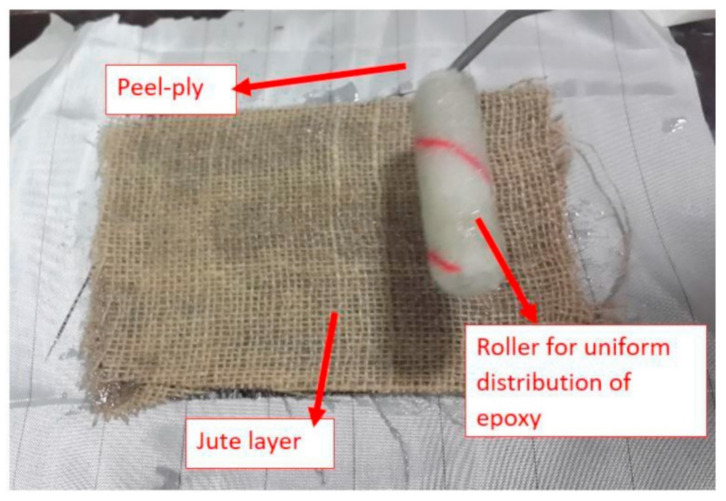
Hand layup method employed for the preparation of composites. Reproduced with permission from Ref. [82].

**Figure 5 polymers-14-03698-f005:**
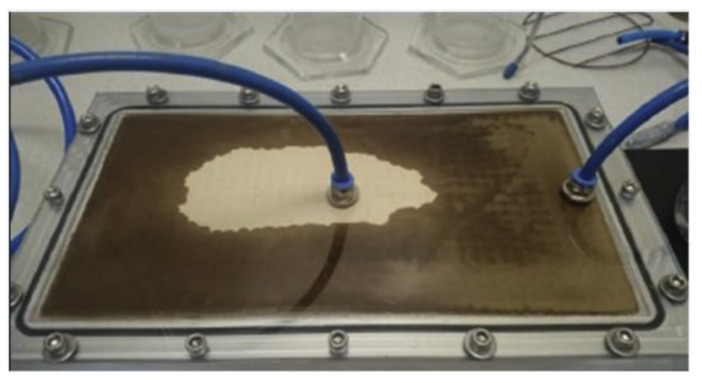
Injection of resin/hardener mixture into fiber plate. Reproduced with permission from Ref. [83].

**Figure 6 polymers-14-03698-f006:**
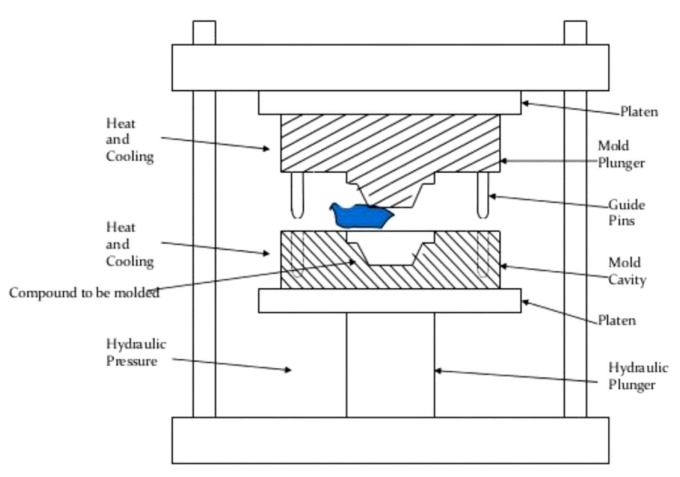
Schematic diagram of the compression molding machine. Reproduced with permission from Ref. [88].

**Figure 7 polymers-14-03698-f007:**
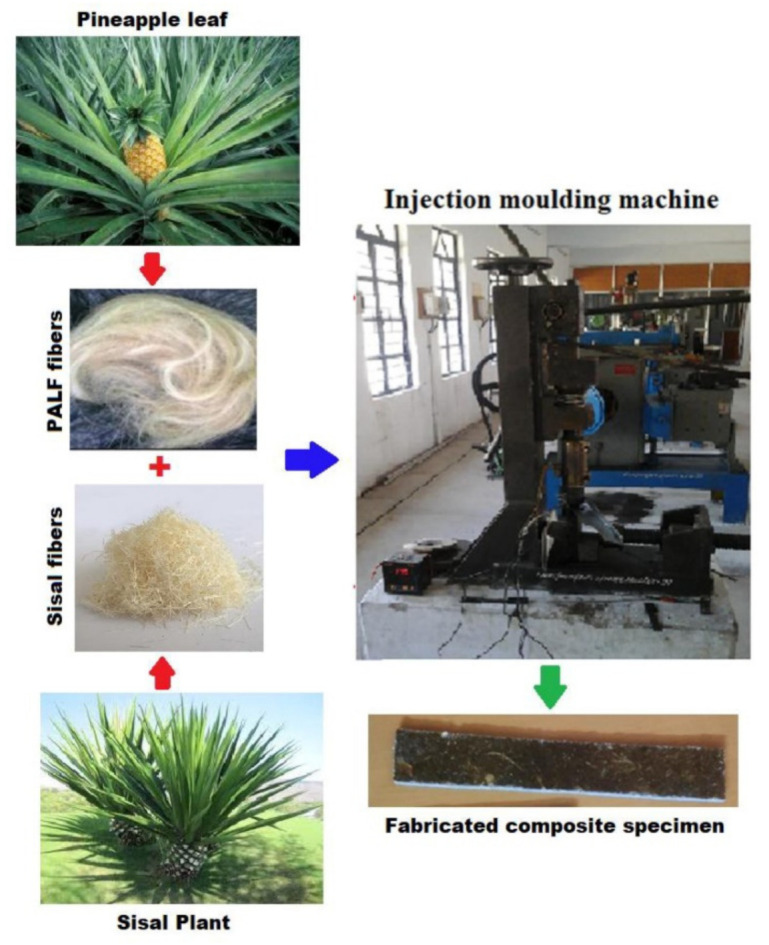
The fabrication process of natural fiber composite specimens. Reproduced with permission from Ref. [89].

**Figure 8 polymers-14-03698-f008:**
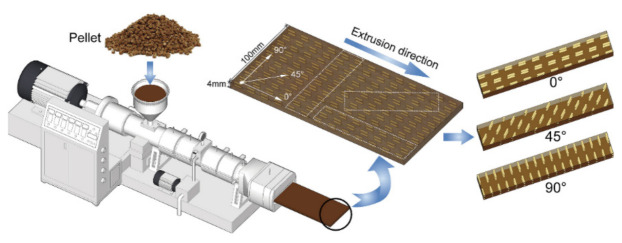
Schematic of the NFRPCs fabrication and sampling direction. Reproduced with permission from Ref. [90].

**Figure 9 polymers-14-03698-f009:**
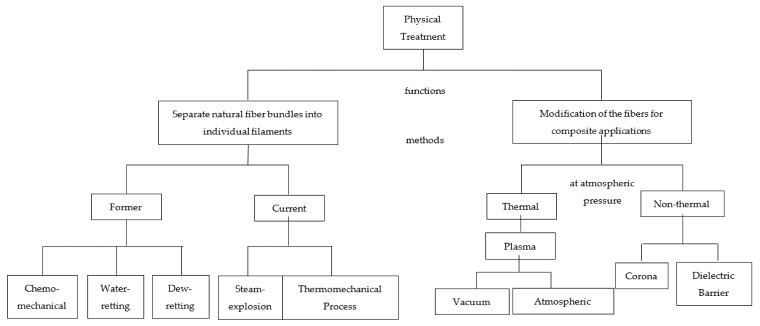
Concept maps of physical treatment on NFRPC [93].

**Figure 10 polymers-14-03698-f010:**
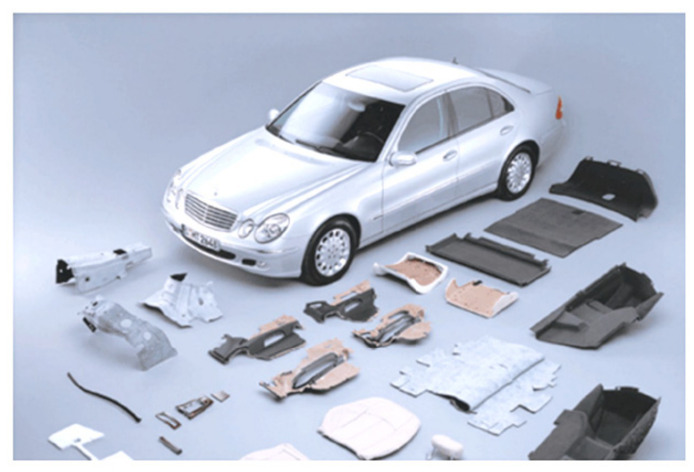
Car components are made up of natural fiber-reinforced composites. Adapted from reference [112] with permission.

**Figure 11 polymers-14-03698-f011:**
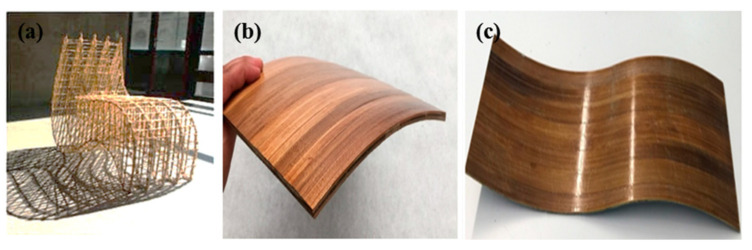
Applications of NFRPCs. (**a**) Hemp chair biocomposites with thermosets binders. Reproduce from reference (Dahy, 2019). Bamboo–PLA composites with (**b**) cylinder concave shape and (**c**) concave–convex shape. Reproduced from reference [115] with permission.

**Figure 12 polymers-14-03698-f012:**
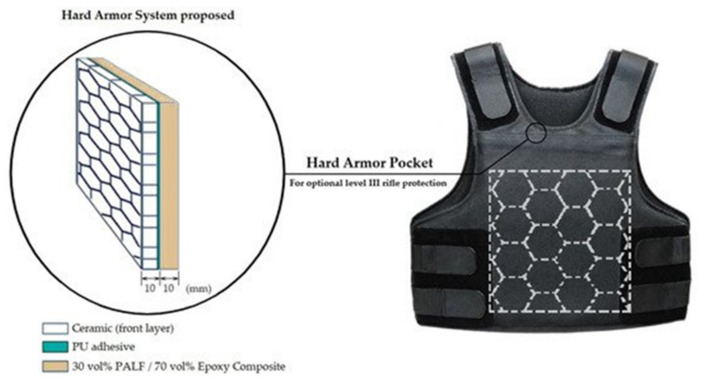
Illustration of hard armor system with natural fiber composites for conventional bulletproof vest. Reproduced from reference [119] with permission.

**Figure 13 polymers-14-03698-f013:**
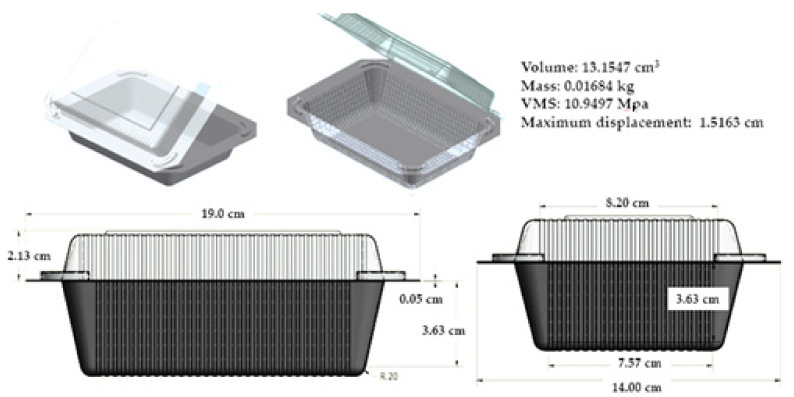
Design representation of the SPF-sago starch food container. Reproduced from reference [124] with permission.

**Figure 14 polymers-14-03698-f014:**
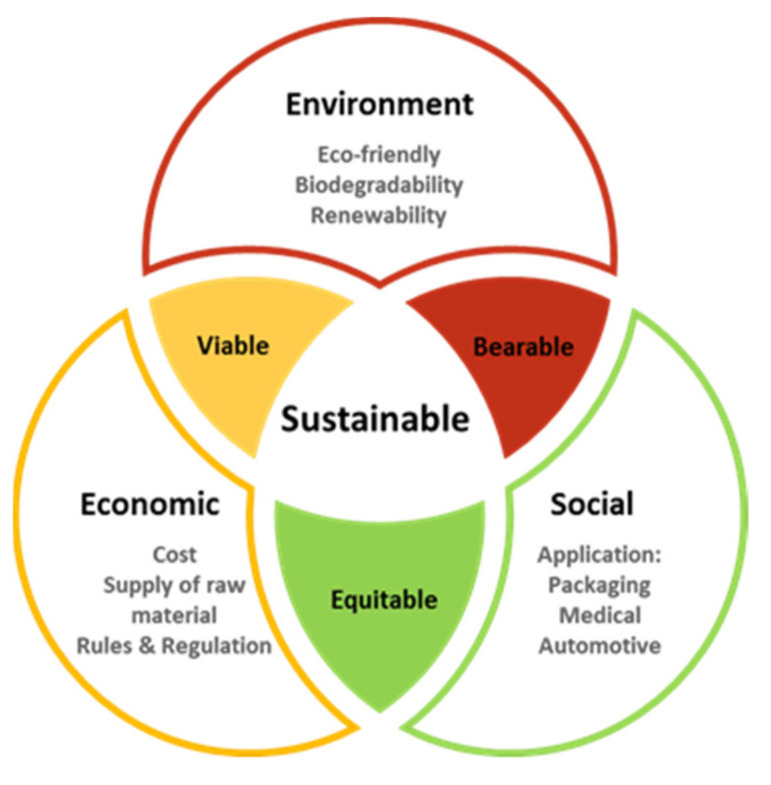
The three cores of sustainability. Adapted from reference [123] with permission.

**Figure 15 polymers-14-03698-f015:**
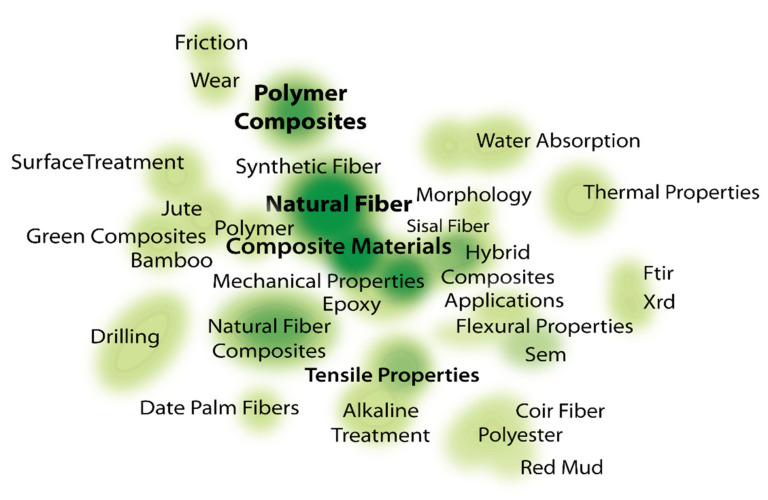
Density visualization of NFRPC.

**Figure 16 polymers-14-03698-f016:**
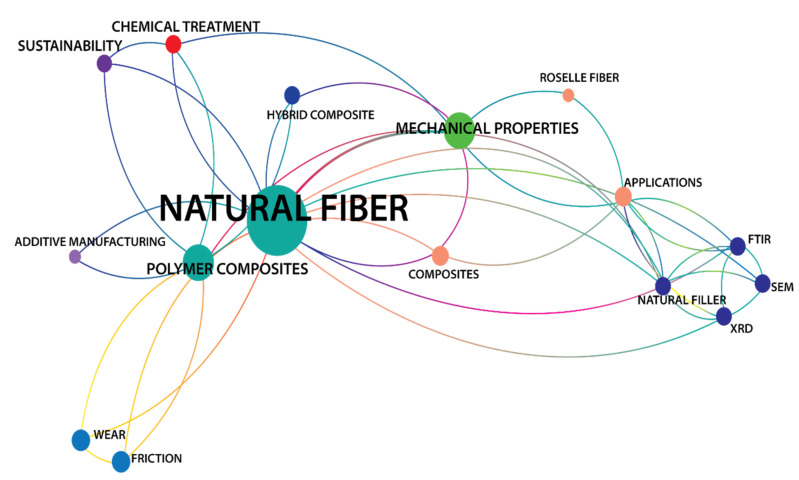
Network visualization of NFRPC.

**Table 1 polymers-14-03698-t001:** Chemical composition of common natural fibers.

Natural Fiber	Chemical Compounds (%)				Ref.
	Cellulose	Hemicellulose	Lignin	Ash	
Coir	36.6	37.0	22.2	1.9	[33]
Spruce	41.6	37.3	19.4	0.4	[33]
Sugar Palm	43.88	10.1	33.24	1.01	[34]
Cornhusk	45.7	35.8	4.03	0.36	[34]
Sugarcane Bagasse	46	24.5	19.5	2.4	[35]
Bamboo	41.8	59.8	29.3	1.5	[36]
Flax	83.3	11.3	2.3	-	[37]
Hemp	55–77	3.7–13	14–22.4	0.8	[37,38]
Jute	45–71.5	13.6–21	12–26	0.5–2.0	[39]
Kenaf	56.81–79.30	9.69–13.59	7.22–18.27	-	[40]
Ramie	68.6–91	5–16.7	0.6–0.7	-	[41,42]
Cotton	82.7–90	5.7	<2	-	[41,42]
Sisal	41.14	41.96	10.40	-	[43]
Pineapple crown	12.93–34.6	25.4–35.49	5.14–26.4	-	[44,45]
Alstonia Scholaris	50.4–68	9.3–10.05	7.7–8.8	1.7–2.1	[46]

**Table 2 polymers-14-03698-t002:** Mechanical properties of natural fibers.

Natural Fiber	Density (g/cm^3^)	Tensile Strength (MPa)	Young’s Modulus (GPa)	Ref.
Bagasse	0.8–1	250–300	17–20	[57]
Ramie	1.4–1.5	400–938	61–128	[58]
Hemp	1.1–1.6	285–1735	14.4–44.0	[58,59]
Kenaf	0.6–1.5	223–1191	11–60	[58,60]
Flax	1.3–1.5	340–1600	25–81	[58,61]
Oil Palm	0.7–1.6	50–400	0.6–9.0	[62]
Jute	1.3–1.5	393–773	13–26.5	[61,63]
Bamboo	1.2–1.5	500–575	27.0–40.0	[64]
Cotton	1.5–1.6	287–800	5.5–12.6	[64]
Sisal	1.3–1.6	468–640	9.4–22	[61,64]
Sugarcane	1.1–1.6	170–350	5.1–6.2	[65]
Coir	1.2–1.6	170–230	3.0–7.0	[66]
Banana	0.5–1.5	711–789	4.0–32.7	[67]

**Table 3 polymers-14-03698-t003:** Chemical Treatment Properties of NFRPC.

Chemical Treatment	Name of the Fiber	Chemical Reagents Used	Method	Structure Improvement	Application	References
Alkaline	Hemp	NaOH	Treated fiber with NaOH at 20 °C for 48 h and washed using distilled water and acetic acid to neutralize the excess of NaOH.	Better fiber-matrix adhesion led to an increase in interfacial energy and thus enhancing the thermal and mechanical properties of the composites	Polymer reinforcements	[69,94]
	Jute,					[69,94]
	Sisal					[69,94]
	Kapok					[69,94]
	Kenaf	NaOH	Treated kenaf fiber with 6% of NaOH solution for 24 h. Then, kenaf fibers were rinsed and immersed into a solution that contained distilled water and 1% acetic acid to neutralize the remaining NaOH. After washing, the kenaf short fibers were dried in an oven for 24 h.	Better physical, morphological, and mechanical properties because of the compatibility of kenaf with polypropylene composites	Automotive	[69,95]
	Napier grass	NaOH	Napier grass fibers were treated with 2% and 5% of NaOH at room temperature for 30 min. The fibers were washed with tap water and distilled water many times and dried at 100 °C.	enhanced tensile properties	Reinforcement for composites	[69,96]
	Carica papaya	NaOH	Carica papaya fibers were treated with 5% NaOH for 60 at 25 °C. Then, the fibers were washed many times using HCI solution and deionized water. Then, the fibers were dried at 100 °C in an oven for moisture removal.	Better performance in mechanical properties, thermal stability, and good interfacial bonding between cellulosic fiber and the matrix.	Light weight industrial	[69,97]
Saline	Sugar palm	Saline	Sugar palm fibers were immersed with 2% saline for 3 h. Then, the fibers were immersed in a mixture of methanol–water (90/10 *w/w*) for 3 h hydrolysis under agitation. The fibers were thoroughly rinsed with distilled water and then oven-dried at 60 °C for 72 h.	Improve properties of sugar palm fiber and enhance fiber-matrix bonding sugar palm fiber–thermoplastic polyurethane composites.	Industrial application	[69,98]
Acetylation	Dombeya buettnerri	Acetyl anhydride	The fibers were soaked with 2% up to 6% of acetyl anhydride for 3 h at room temperature. Then, the fibers were washed with tap water and repeatedly rinsed with distilled water until all excess acid had been removed. Then, the fibers were dried for 2 h at 105 °C.	Enhanced surface morphology and mechanical properties.	Engineering materials applications	[69,99]
	Combretum racemosum					
	Banana (Musa parasidica)					
Alkaline hydrogen peroxide	Citrus fibers	Hydrogen peroxide	The citrus fibers were immersed in hydrogen peroxide for 4 h at 60 °C. Then, the fibers were adjusted to pH 6 with acid hydrochloric (1.0 M) at 25 °C. The mixture was centrifuged at 6000× *g* for 15 min; the residue was then collected and washed in pure ethanol and dried via oven at 60 °C for 7 h.	High water holding and swelling capacities could be used as emulsifiers in juice and jam. It also has better thermal stability and viscosity properties.	Application in food industry	[100]
Benzoylation	Sisal fiber,	Benzoyl chloride		Increase strength of composite and thermal stability, decrease water absorption	Industrial application	[101]
	Sugar palm		Soaked with a mixture solution of 1% NaOH and 5 mL of C_7_H_5_ClO with respective soaking times. Then, fibers were washed and soaked in absolute ethanol for 1 h, washed again until pH became neutral, and dried overnight at 50 °C.	Improvement in tensile strength	Furniture and components inside vehicle	[102]
Acrylation and Acrylonitrile Grafting	Flax-fiber	Acrylic acid solution	Flax fibers were immersed in NaOH solution for 0.5 h and then soaked in acrylic acid solution at 50 °C for 1 h, washed with distilled water, and dried.	Improving the physical and mechanical properties	Plastic, automobile, and packaging industries	[103]
Maleated Coupling Gents	Jute fiber	Maleic anhydride- polypropylene (MAPP)	The fibers were immersed in MAPP solution in toluene at 100 °C.	Increase in mechanical strength	Industrial applications with offer cost-effective and value-added composite material	[104]
Permanganate Treatment	Sisal fiber	Potassium permanganate KmnO_4_	Sisal fibers were soaked carefully in a solution of KmnO_4_/acetone with a concentration of 0.02% for 3 min. After that, the fibers were taken out, washed many times with distilled water, and dried	Improve fiber strength and fiber-matrix adhesion	Industrial application	[101]
Peroxide Treatment	Sisal fibers	Benzoyl peroxide from acetone solution	Fibers were coated with benzoyl peroxide from acetone solution after alkali pre-treatment. A saturated solution of the peroxide in acetone was used. Fibers were then dried.	Enhance in tensile properties	Substitute the wood	[105]
Isocyanate Treatment	Pineapple leaf fiber	toluene solution containing poly(methylene)-poly(phenyl)isocyanate	Fibers were dipped in toluene solution containing PMPPIC (5 wt% of fiber) for half an hour at 50 °C. The fibers were then decanted and dried in an air oven at 70 °C for 2 h. Later these were mixed with polyethylene using toluene as the solvent containing PMPPIC (6 wt% of fiber) at a temperature of 120 °C.	Enhance mechanical properties	Structural and non-structural application	[106]
Ionic Liquid	Chitin fiber	1-ethyl-3-methylimidazolium acetate	Chitin derived from shrimp shell biomass that has been thermally pretreated, pressed, and ground. Chitin was isolated using a microwave-assisted dissolution of [C2mim][OAc], followed by water coagulation, washing, and oven drying.	Improve the mechanical strength of chitin fibers	High-performance chitinous sorbents for applications such as metal recovery from seawater	[107]
Thermal decomposition kinetic	Wood, bamboo, agricultural residue, and bast fibers	Phosphonium ionic liquids	All raw materials were washed with water to remove impurities before being dried in an oven at 75 degrees Celsius for 12 h. The dried materials were then ground and screened using a Wiley mill. For testing, samples with particle sizes ranging from 20 to 28 meshes were collected. Various degradation models, including the Kissinger, Friedman, Flynn–Wall–Ozawa, and modified Coats–Redfern methods to determine the apparent activation energy of these fibers.	Improve the thermal stability of the fibers	Renewable biomass energy/natural fuels and forest fire propagation control, practical engineering applications.	[108]

**Table 4 polymers-14-03698-t004:** Physical and Chemical Treatment for Surface Modifications [93].

Type of Treatment	Name of Treatment	Mechanism of Treatment	Improvement
Physical	Corona	The formation of a high-energy electromagnetic field close to charged thin wires/points induced ionization species (ions, radicals, etc.) and activated for surface modification through introduction of oxygen-containing functional groups	- Existence of hydroperoxide groups that could initiate grafting of the matrix chains led to significant improvement of interfacial shear strength
	Plasma	Similar mechanism to corona. However, the apparatus required a vacuum chamber and gas feed to maintain the appropriate composition of the gaseous mixture.	
	Mercerisation	Soaking the fiber in sodium hydroxide.	- Improves adhesive characteristics by removing natural and artificial impurities and promotes rough surface topography Fiber fibrillation (breaking down the composites fiber bundle into smaller fiber) Increase the effective surface area available for contact with the wet matrix Enhances the reactivity
	Heat treatment	Heated and the fiber undergoes physical (enthalpy, weight, strength, color, and crystallinity) and chemical changes (reduction degree of polymerization by bond scission, creation of free radicals, formation of carbonyl, carboxyl, and peroxide groups)	- Increased yield strength
Chemical	Esterification-based treatments	- Use of a variety of chemicals to form ester bonds with the fiber surface To coat the OH groups (hydrophilic character) with molecules that have a more hydrophobic nature Chemical process used for esterification: acetylation, benzylation, propionylation, and treatment with stearate	- Remove non-crystalline constituents of the fibers, thus altering the fiber surface topography
		$Fiber\;–\;OH+CH3\;–\;C\left(\;\equiv O\right)-O-C\left(\equiv O\right)-CH3\;\to \;Fiber-OCOCH3+CH3\;COOH(Acetylation$) $\mathrm{React}\;\mathrm{the}\;\mathrm{hydroxyl}\;\mathrm{group}\;\left(-OH\right)$ of the fiber constituents with acetyl groups (CH3 CO−) for full esterification	- Modifying surface of natural fibers and making it more hydrophobics Reducing swelling of wood in water Reducing moisture absorption Enhanced thermal stability
		- $Fiber\;–\;OH+NaOH\;\to \;Fiber-O^{-}Na^{+}+H2O\;\left(Benzylation\right)$ Two stages. Firstly, immersed in sodium hydroxide solution and preceded with benzylchloride	- Promotes compatibility with polymers containing aromatic rings
		Propionylation - Similar method to acetylation; only had one more methyl group than the acetic anhydride	- Interface stress transfer efficiency improved
		Treatment with stearate $\mathrm{Modified}\;\mathrm{fiber}\;\mathrm{with}\;\mathrm{stearic}\;\mathrm{acid}\;\left(C17H35\;COOH\right)$	- Formed stable ester bonds with the hydroxyl group
	Saline coupling agents	- Formation of covalent bonds between the Y group and the matrix during curing. $\;\;\;\;\;\;\;\;\;\;\;\;\;\;\;\;\;\;\;\;\;\;\;\;\;\;\;\;\;\;\;\;\;\;\;\;\;\;\;\;\;\;\;\;\;\;CH2\;CHSi\left(OC2H5\right)3\overset{H2O}{\to }CH2\;CHSi\left(OH\right)3\;+3C2H5\;OH$ $\;\;\;\;\;\;\;\;\;\;\;\;\;\;\;\;\;\;\;\;\;\;\;CH2\;CHSi\left(OH\right)3\;$ $+Fiber-OH\to \;CH2\;CHSi\left(OH\right)2O-Fiber+H2O$	- Reduce the number of hydroxyl groups on the surface of polar materials such as natural fibers rendering them more hydrophobic
	Graft copolymerization	- Two different mechanisms are involved; polymerization on the fiber surface by free radical and free radical formed by cracking the cellulose molecules Graft copolymerization can be divided into three subcategories; treatment with triazine coupling agents, treatment with isocyanates, and treatment with maleic anhydride	
		- Triazine coupling agents; treated with three derivatives of trichloro-s-triazine (2-octyloamino 4, 6-dichloro-s-triazine, methacrylic aci, 3-(4,6–dichloro-s-triazine-2-yl) aminopropyl ester, 2-diallylamino 4,6–dichloro-s-triazine)	- Tensile strength increased
		- Isocyanate; formation of covalent bonds between cellulose and isocyanate coupling agent, which hydrophobises the fiber surface Increased interaction between polymethylene (polyphenylene) isocyanate (PMPPIC) and polystyrene that contain benzene ring due to interaction of delocalized π− electrons of the benzene rings (Van der Waals type of interactions) of both polymers	- Increase in stress and modulus values of the composites superior mechanical properties and dimensional stability
		- Maleic anhydride; chemical bonds of esoteric nature, as well as hydrogen bonds, are formed between the maleic anhydride functional groups of polypropylenes and the hydroxyl group of cellulose.	- Increased in tensile strength and Young’s modulus
Various chemical	Dimethylurea (DMU)	- Reaction DMU with OH group of the fibers that subsequently almost eliminated any fiber–fiber interaction resulting from intermolecular hydrogen bonds.	- Tensile modulus and elongation increased Better dispersion of flax fibers in the matrix
	Phenol formaldehyde (PF)	- Methylol groups react with hydroxyl groups, forming stable ether bonds, while at the same time, it contains hydrophobic polymer chains.	- Water uptake of composites decreases, and moisture content of treated fiber composites is 50% lower than non-treated fiber composite

**Table 5 polymers-14-03698-t005:** NFRPC application in various industrial applications.

Authors	Descriptions	Applications
1. [141]	Study on the mechanical properties of epoxy composites reinforced by jute ramie hybridization. The hybrid composites with the desired and preferable properties can be manufactured using a hand-lay-up technique and used in various industrial applications.	Various industrial applications
2. [139]	A comprehensive study on the drilling behavior of different compositions of Polypropylene composites and Polyethylene composites.	Various industrial applications
3. [149]	Study on the extraction, processing, properties, and application of natural fiber-reinforced composites derived from leaves, namely pineapple, sisal, and abaca.	Various industrial applications
4. [144]	Addressing the natural fiber reinforced hybrid nanocomposite manufactured by the incorporation of high-frequency microwave treated Plantain (Musa paradisiaca) fiber and multiwalled carbon nanotubes (MWCNT) using a single epoxy resin matrix.	Automobile
5. [142]	Addressing the challenges and opportunities associated with the use of natural fiber–reinforced polymer composites in the automotive and furniture industry.	Automotive and furniture
6. [150]	This study focuses on a comparative experimental analysis of the effects of conventional drilling (CD) and a hybrid ultrasonically assisted drilling (UAD) of hemp fiber–reinforced vinyl ester composite laminate.	Various industrial applications
7. [151]	This book highlights a totally new research theme in biopolymer-based composite materials and bioenergy.	Various industrial applications
8. [148]	Study on the physical and thermal properties such as density, water absorption, thermal conductivity, specific heat, and thermal diffusivity for short fiber–reinforced hybrid composites.	Building components and automobiles
9. [143]	Overview of the polymeric materials recycling, as well as the main challenges in obtaining natural fiber–reinforced polymer composites.	Automotive
10. [152]	Study on the natural fiber reinforced polymer composite materials from coconut fibers for fiberglass boat building.	Boats
11. [153]	Development of high-performance materials made from coconut fiber to replace the industrial core and foam. It is used to increase the thickness of the fiberglass boat.	Boats
12. [154]	Addressing the composite material, which is to be incorporated in replacing the conventional steel leaf spring and utilizing the fiber, which poses a threat to the environment.	Various industrial applications
13. [145]	Overview of the developments of natural fibers reinforced composites, in terms of their physical and mechanical properties and crashworthiness properties.	Aerospace and automotive
14. [140]	A critical review of the most recent development of natural fiber for construction (ceiling paneling, partition boards) and transportation (automobiles, railway coaches, aerospace) industries.	Construction and transportation
15. [155]	Review article on fiber reinforced composites as cheaper construction and building material.	Various industrial applications
16. [47]	Present a model to evaluate the available polymers for natural fibers to enhance the industrial sustainability theme. Polymer evaluations are illustrated for different technical criteria in order to facilitate the polymer selection process for various industrial applications with high confidence levels.	Various industrial applications
17. [156]	Study on the mechanical behavior of natural fiber reinforced composite panels. The present work includes the characterization and modeling of jute and coir fiber-reinforced hybrid composite materials.	Various industrial applications
18. [47]	Study on the feasibility of using the date palm fibers in the natural fiber reinforced polymer composites (NFC) for the automotive industry. This adoption has a significant environmental influence on achieving an efficient, sustainable waste management practice.	Automotive
19. [146]	Presents the free vibration characteristics of newly identified Phoenix Sp fiber reinforced polymer matrix composite beams and determines the physical, chemical, and mechanical properties of the fiber.	Automobile and aerospace
20. [147]	Present the theory of sorption of liquids into porous textile structures and the results of a computer simulation of liquid absorption and transport into a nonwoven textile structure used for baby diapers.	Textile

## Data Availability

The data that support the findings of this study are available on request from the corresponding author.

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
