# Peer review of "A Review on Natural Fiber Reinforced Polymer Composites (NFRPC) for Sustainable Industrial Applications"

_polymers, 2022, doi:10.3390/polym14173698_

Round 1

Reviewer 1 Report

In this manuscript, authors stated an overview about a meaningful technology of natural fiber reinforced polymer composites (NFRPC), which have promising potential in reducing environmental pollution and improving surface functionalities. Through addressing the basic information about NFRPCs of various kinds, physiochemical properties, fabrication, technique process and modifications treatment, the concept of NFRPCs have successfully established. The content of NFPRC for sustainable industrial application and prospects in industry indicate the development space of this technology. Actually, this manuscript is well written. However, there are still some issues to be addressed. This reviewer would suggest a moderate revision before its acceptance.

1.       There are many similar review articles regarding the natural fiber reinforced polymer composites, such as the following two similar review articles: Natural fiber reinforced polymer composites in industrial applications: feasibility of date palm fibers for sustainable automotive industry; An empirical review of the recent advances in treatment of natural fibers for reinforced plastic composites. Authors should modify their review article to further show the novelty of this work.

2.       Ionic liquids are reported to treat the natural fibers, which should be mentioned in the manuscript. Please refer: Processing and valorization of cellulose, lignin and lignocellulose using ionic liquids; Ma Chunhui, Sum Jinde, Li Wei, Luo Sha, Liu Shouxin. Application progress of ionic liquids in the field of lignin depolymerization.Journal of Forestry Engineering,2021,6(05):14-26.doi:10.13360/j.issn.2096-1359.202008012; etc.

3.       The layout and size of images and tables should be modified.

4.       The thermal stability of the natural fibers could be clarified with supporting articles: Thermal decomposition kinetics of natural fibers: activation energy with dynamic thermogravimetric analysis.

5.       The texts in some images should be adjusted to have a better readability.

6.       The recyclable of NFRPCs could be further illustrated clearly.

7.       Sugarcane bagasse fiber, hyacinth fibers and bamboo fibers have also been reported for the preparation of fiber reinforced plastic composites. More comprehensive literature searching should be performed: Development and characterization of food packaging bioplastic film from cocoa pod husk cellulose incorporated with sugarcane bagasse fibre; Zhao He, Miao Qingxian, Huang Liulian, Zhou Xiaxing, Chen Lihui. Preparation of long bamboo fiber and its reinforced polypropylene membrane composites. Journal of Forestry Engineering,2021,6(05):96-103.doi:10.13360/j.issn.2096-1359.202101020Packaging and degradability properties of polyvinyl alcohol/gelatin nanocomposite films filled water hyacinth cellulose nanocrystals; etc.

8.       The serial number of images corresponds to the error. There are some mistakes in this manuscript. First, second to last paragraph on page 28: ”Figure 7 exhibits some examples of the commercialized car components made up of natural fiber reinforced composites”, “figure 7” may be replaced with “figure 23”; second, the third paragraph on page 30:” Figure 9 shows the illustration of hard armor system with NFRPC for bulletproof vest as proposed in their study”, “figure 9” may be replaced with “figure 25”; third, the fourth paragraph on page 31:” The actual concept de-sign of the container is displayed on Figure 10”, “figure 10” may be replaced with “figure 26”; fourth, the third paragraph on page 33:” The concept of sustainability is based on three main cores which consist of environmental, social and economic point of view as illustrated in Figure 11”, “figure 11” may be replaced with “figure 27”.

9.       The serial number of tables is wrong. The table number of ”NFRPC application in various industrial applications” may be 5.

10.   The logic of the fig.28 and fig.29 should be further modified to have a better understanding for readers.

11.   Bamboo fibers are one kinds of important natural fibers as reinforcements. It is important to present the preparation, modification and functionizaiton of bamboo fibers with supporting article: Chen Lihui, Cao Shilin, Huang Liulian, Wu Hui, Hu Huichao, Liu Kai, Lin Shan. Development of bamboo cellulose preparation and its functionalization. Journal of Forestry Engineering,2021,6(04):1-13.doi:10.13360/j.issn.2096-1359.202104011

12.     One image is suggested at the end of introduction to summarize the whole contents of this review for a better understanding of readers. Another image is suggested to highlight the challenges and solutions for guiding the future research.

13.     There are many errors in the format of references. Authors should carefully recheck the whole reference list to make sure full information is provided, especially the page numbers.

14.     Please delete (XXXX) in refer. 73.

15.     It is suggested to remove the SEM images from Table 3, which occupy too much spaces and do not provide too much useful information.

Author Response

Manuscript ID: polymers-1865885

Response to Reviewers

Dear Ms. Cori Jia, Thank you for giving us the opportunity to submit a revised draft of the manuscript “A Review on Natural Fiber Reinforced Polymer Composites (NFRPC) for Sustainable Industrial Applications” for publication in the Polymers journal. We appreciate the time and effort that you and the reviewers dedicated to providing feedback on our manuscript and are grateful for the insightful comments on and valuable improvements to our paper. We have incorporated most of the suggestions made by the reviewers. Those changes are highlighted within the manuscript. Please see below, in blue, for a point-by-point response to the reviewers’ comments and concerns. All page numbers refer to the revised manuscript file with tracked changes.

Reviewers' Comments to the Authors:

Reviewer 1

Comments and Suggestions for Authors

In this manuscript, authors stated an overview about a meaningful technology of natural fiber reinforced polymer composites (NFRPC), which have promising potential in reducing environmental pollution and improving surface functionalities. Through addressing the basic information about NFRPCs of various kinds, physiochemical properties, fabrication, technique process and modifications treatment, the concept of NFRPCs have successfully established. The content of NFPRC for sustainable industrial application and prospects in industry indicate the development space of this technology. Actually, this manuscript is well written. However, there are still some issues to be addressed. This reviewer would suggest a moderate revision before its acceptance.

Author response: Thank you. We are grateful to the reviewer for his/her insightful comments on our paper. We have highlighted the changes within the manuscript. Here is a point-by-point response to the reviewers' comments and concerns.

  1. There are many similar review articles regarding the natural fiber reinforced polymer composites, such as the following two similar review articles: Natural fiber reinforced polymer composites in industrial applications: feasibility of date palm fibers for sustainable automotive industry; An empirical review of the recent advances in treatment of natural fibers for reinforced plastic composites. Authors should modify their review article to further show the novelty of this work.

Author response: Thank you for bringing this to our attention. Although there are many similar review articles on natural fibre reinforced polymer composites, such as those mentioned above, our review paper focuses on a variety of natural fibres types available rather than just one type of natural fibre, such as date palm fibres. Furthermore, our paper highlighted several types of applications, including automotive, military, aerospace, building and constructions, rather than focusing solely on one aspect of application in automotive industry, which was previously published. Moreover, several types of treatments, including physical and chemical treatments of natural fibre, were highlighted in our paper. In addition, our paper had highlighted important subtopics such as "Cost effectiveness of NFRPCs in sustainable industrial applications" and "Roadmap of NFRPC to Moving into Industry 4.0," emphasising the uniqueness of our work.

  1. Ionic liquids are reported to treat the natural fibers, which should be mentioned in the manuscript. Please refer: Processing and valorization of cellulose, lignin and lignocellulose using ionic liquids; Ma Chunhui, Sum Jinde, Li Wei, Luo Sha, Liu Shouxin. Application progress of ionic liquids in the field of lignin depolymerization.Journal of Forestry Engineering,2021,6(05):14-26.doi:10.13360/j.issn.2096-1359.202008012; etc.

Author response: As suggested by the reviewer, we have added the information regarding the ionic liquids treatment to treat natural fibers which had been mentioned in the Table 3.

  1. The layout and size of images and tables should be modified.

Author response: We agree with the reviewer’s assessment. Accordingly, throughout the manuscript, we have modified the layout and size of images and tables as per suggested.

  1. The thermal stability of the natural fibers could be clarified with supporting articles: Thermal decomposition kinetics of natural fibers: activation energy with dynamic thermogravimetric analysis.

Author response: Thank you for this suggestion. It would have been interesting to explore this aspect.

Supporting articles regarding thermal stability of natural fibers had been clarified with supporting articles which had been mentioned in the Table 3.

  1. The texts in some images should be adjusted to have a better readability.

Author response: Thank you for pointing this out. The texts in some images had been adjusted to have a better readability.

  1. The recyclable of NFRPCs could be further illustrated clearly.

Author response: We think this is an excellent suggestion. We have added the image of sustainability of NFRPC which had been illustrated clearly in Figure 1.

  1. Sugarcane bagasse fiber, hyacinth fibers and bamboo fibers have also been reported for the preparation of fiber reinforced plastic composites. More comprehensive literature searching should be performed: Development and characterization of food packaging bioplastic film from cocoa pod husk cellulose incorporated with sugarcane bagasse fibre; Zhao He, Miao Qingxian, Huang Liulian, Zhou Xiaxing, Chen Lihui. Preparation of long bamboo fiber and its reinforced polypropylene membrane composites. Journal of Forestry Engineering,2021,6(05):96-103.doi:10.13360/j.issn.2096-1359.202101020Packaging and degradability properties of polyvinyl alcohol/gelatin nanocomposite films filled water hyacinth cellulose nanocrystals; etc.

Author response: We think this is an excellent suggestion. To address the reviewer’s comment, we have revised the manuscript and added the comprehensive literature in the Section 2.2 Fabrication and Technique Process of NFRPC.

  1. The serial number of images corresponds to the error. There are some mistakes in this manuscript. First, second to last paragraph on page 28: ”Figure 7 exhibits some examples of the commercialized car components made up of natural fiber reinforced composites”, “figure 7” may be replaced with “figure 23”; second, the third paragraph on page 30:” Figure 9 shows the illustration of hard armor system with NFRPC for bulletproof vest as proposed in their study”, “figure 9” may be replaced with “figure 25”; third, the fourth paragraph on page 31:” The actual concept de-sign of the container is displayed on Figure 10”, “figure 10” may be replaced with “figure 26”; fourth, the third paragraph on page 33:” The concept of sustainability is based on three main cores which consist of environmental, social and economic point of view as illustrated in Figure 11”, “figure 11” may be replaced with “figure 27”.

Author response: We thank the reviewer for their comment. To address the reviewer’s comment, we have revised the manuscript, all figures had been revised accordingly in the manuscript.

  1. The serial number of tables is wrong. The table number of ”NFRPC application in various industrial applications” may be 5.

Author response: We thank the reviewer for their comment. To address the reviewer’s comment, we have revised the manuscript, and replaced Table 3 with Table 5.

  1. The logic of the fig.28 and fig.29 should be further modified to have a better understanding for readers.

Author response: We think this is an excellent suggestion. We have further modified the logic of the Figure 15. Density visualization of NFRPC and Figure 16. Network visualization of NFRPC for a better understanding for readers.

  1. Bamboo fibers are one kinds of important natural fibers as reinforcements. It is important to present the preparation, modification and functionizaiton of bamboo fibers with supporting article: Chen Lihui, Cao Shilin, Huang Liulian, Wu Hui, Hu Huichao, Liu Kai, Lin Shan. Development of bamboo cellulose preparation and its functionalization. Journal of Forestry Engineering,2021,6(04):1-13.doi:10.13360/j.issn.2096-1359.202104011

Author response: We think this is an excellent suggestion. We have revised the manuscript and have further added the comprehensive review regarding the preparation, modification and functionalization of bamboo fibers in Section 2.2 Fabrication and Technique Process of NFRPC.

  1. One image is suggested at the end of introduction to summarize the whole contents of this review for a better understanding of readers. Another image is suggested to highlight the challenges and solutions for guiding the future research.

Author response: We think this is an excellent suggestion. We have provided one image, Figure 1. Sustainability of NFRPC at the end of introduction to summarize the whole contents of this review for a better understanding of readers. Another image, Figure 2. Challenges faced by non-renewable materials and solutions offered by NFRPC has also been added to highlight the challenges and solutions for guiding the future research.

  1. There are many errors in the format of references. Authors should carefully recheck the whole reference list to make sure full information is provided, especially the page numbers.

Author response: We thank the reviewer for their comment. To address the reviewer’s comment, we have revised the manuscript, all references had been revised accordingly in the manuscript.

  1. Please delete (XXXX) in refer. 73.

Author response: We thank the reviewer for their comment. To address the reviewer’s comment, we have deleted (XXXX) in the References section for reference Getu, D., Nallamothu, R. B., Masresha, M., Nallamothu, S. K., & Nallamothu, A. K. (2020). Production and characterization of bamboo and sisal fiber reinforced hybrid composite for interior automotive body application. Materials Today: Proceedings, 38, 2853–2860. https://doi.org/10.1016/j.matpr.2020.08.780

  1. It is suggested to remove the SEM images from Table 3, which occupy too much spaces and do not provide too much useful information.

Author response: We think this is an excellent suggestion. All SEM images from Table 3 had been removed.

Reviewer 2 Report

Reviewer’ comments

 The paper presents an overview of the technological challenges, processing techniques, characterization, properties, and potential applications of NFRPC for sustainable industrial applications. It provides an in-depth look at the properties and the benefits of natural fiber in many fields such as Aerospace, Automotive, medical, Building constructions, sports, etc…

This topic is of great importance in the structural field.

The subject of this paper is coherent with the scope of the Polymers journal and the article adheres to the journal's standards.

The title clearly describes the content found in the article.

The abstract gives good understanding of the article.

The experimental program and procedure followed are explained clearly.

The results are shown in a logical way.

The article is worthy to be published - at the same time, minor revision is required to the authors: 

1    1To facilitate the reader insert in the text a list of acronyms with their respective meanings.

 Authors are invited to rearrange the section “Conclusions”. In the revised version resubmitted, the conclusions have to be sound and justified and follow logically the various aspects dealt with.

3    Please, check the References section. In bibliographical references, it is advisable to mention articles of the same publisher, and articles in the constructions fields, for example it is suggested: https://doi.org/10.4028/www.scientific.net/amr.778.537

Author Response

Reviewer 2

The paper presents an overview of the technological challenges, processing techniques, characterization, properties, and potential applications of NFRPC for sustainable industrial applications. It provides an in-depth look at the properties and the benefits of natural fiber in many fields such as Aerospace, Automotive, medical, Building constructions, sports, etc…

This topic is of great importance in the structural field.

The subject of this paper is coherent with the scope of the Polymers journal and the article adheres to the journal's standards.

The title clearly describes the content found in the article.

The abstract gives good understanding of the article.

The experimental program and procedure followed are explained clearly.

The results are shown in a logical way.

The article is worthy to be published - at the same time, minor revision is required to the authors: Author response: Author response: Thank you. We are grateful to the reviewer for his/her insightful comments on our paper. We have highlighted the changes within the manuscript. Here is a point-by-point response to the reviewers' comments and concerns.

  1. To facilitate the reader insert in the text a list of acronyms with their respective meanings.

Author response: Thank you for pointing this out. We have added the list of acronyms with their respective meanings in our manuscript.

  1. Authors are invited to rearrange the section “Conclusions”. In the revised version resubmitted, the conclusions have to be sound and justified and follow logically the various aspects dealt with.

Author response: We have added the suggested content to the manuscript on the section Conclusions.

3    Please, check the References section. In bibliographical references, it is advisable to mention articles of the same publisher, and articles in the constructions fields, for example it is suggested: https://doi.org/10.4028/www.scientific.net/amr.778.537

Author response: Thank you for pointing this out. All reference in the References section has been carefully updated.

Round 2

Reviewer 1 Report

Authors have addressed most of the issues well.

1. The figure 15 should be further modified with bigger texts.

2. The characterization methods in Figure 16 should be written with all letters in capital style.

3. Authors made responses to previous comments 2 and comments 11, however, forgot to mention the important suggested articles in the manuscript.

4. More perspectives on the challenges and future possible solutions should be provided.

Author Response

  1. The figure 15 should be further modified with bigger texts

Author response: We thank the reviewer for their comment. To address the reviewer’s comment, we have revised the figure 15 and modified with bigger texts.

  1. The characterization methods in Figure 16 should be written with all letters in capital style.

Author response: We thank the reviewer for their comment. To address the reviewer’s comment, we have revised the figure 16 and modified with all letters in capital style.

  1. Authors made responses to previous comments 2 and comments 11, however, forgot to mention the important suggested articles in the manuscript.

Author response: We thank the reviewer for their comment. To address the reviewer’s comment, we have added the important suggested articles in the manuscript in Section 2.2 Fabrication and Technique Process of NFRPC, highlighted with yellow in colour, as well as updating the references in the References section.

  1. More perspectives on the challenges and future possible solutions should be provided.

Author response: We thank the reviewer for their comment. To address the reviewer’s comment, we have added more perspectives on the challenges and future possible solutions in Section 5. Challenges and Future Perspectives, highlighted with yellow in colour.
